# Actin dynamics tune the integrated stress response by regulating eukaryotic initiation factor 2α dephosphorylation

Joseph E Chambers[1,2†], Lucy E Dalton[1,2†], Hanna J Clarke[2,3], Elke Malzer[1,2], Caia S Dominicus[1,2], Vruti Patel[1,2], Greg Moorhead[4], David Ron[2,3], Stefan J Marciniak[2,3]*

[1]Cambridge Institute for Medical Research, Department of Medicine, University of Cambridge, Cambridge, United Kingdom; [2]Wellcome Trust MRC Building, University of Cambridge, Cambridge, United Kingdom; [3]Cambridge Institute for Medical Research, University of Cambridge, Cambridge, United Kingdom; [4]Department of Biological Sciences, University of Calgary, Calgary, Canada

**Abstract** Four stress-sensing kinases phosphorylate the alpha subunit of eukaryotic translation initiation factor 2 (eIF2α) to activate the integrated stress response (ISR). In animals, the ISR is antagonised by selective eIF2α phosphatases comprising a catalytic protein phosphatase 1 (PP1) subunit in complex with a PPP1R15-type regulatory subunit. An unbiased search for additional conserved components of the PPP1R15-PP1 phosphatase identified monomeric G-actin. Like PP1, G-actin associated with the functional core of PPP1R15 family members and G-actin depletion, by the marine toxin jasplakinolide, destabilised the endogenous PPP1R15A-PP1 complex. The abundance of the ternary PPP1R15-PP1-G-actin complex was responsive to global changes in the polymeric status of actin, as was its eIF2α-directed phosphatase activity, while localised G-actin depletion at sites enriched for PPP1R15 enhanced eIF2α phosphorylation and the downstream ISR. G-actin's role as a stabilizer of the PPP1R15-containing holophosphatase provides a mechanism for integrating signals regulating actin dynamics with stresses that trigger the ISR.

*For correspondence: sjm20@ cam.ac.uk

†These authors contributed equally to this work

## Introduction

In eukaryotes, regulation of protein biosynthesis defends against proteotoxic stress and balances anabolic growth with nutrient availability (*Jackson et al., 2010*). The eukaryotic translation initiation factor 2 complex recruits the initiator methionyl-tRNA to ribosomes in a GTP-dependent catalytic cycle, but phosphorylation of its alpha subunit (eIF2α) by a family of stress-sensing kinases inhibits guanine nucleotide exchange, attenuating translation initiation and with it, global protein synthesis (*Clemens, 1996*; *Jackson et al., 2010*). During endoplasmic reticulum stress, the eIF2α kinase PERK triggers attenuation of protein synthesis (*Harding et al., 1999*), while other members of the family respond variously to amino acid deprivation, viral infection or heme deficiency (*van 't Wout et al., 2014*). This ability of eIF2α phosphorylation to integrate signals from multiple, apparently unrelated, stresses led this pathway to be named the 'integrated stress response' (ISR) (*Harding et al., 2003*). Attenuated global protein synthesis is cytoprotective early in the stress response. The ISR also involves the translational induction of the transcription factors ATF4 and CHOP, which activate a transcriptional programme that has pro-survival effects in the short term, whilst prolonged activation leads to a switch to apoptosis (*Zinszner et al., 1998*; *Marciniak et al., 2004*; *Lu et al., 2014*).

In metazoans, inactivation of the ISR is mediated by a catalytic, protein phosphatase 1 (PP1) subunit in complex with a regulatory subunit (PPP1R15) responsible for targeting eIF2α (*He et al., 1998*;

**eLife digest** For a cell to build a protein, it must first copy the instructions contained within a gene. A complex molecular machine called a ribosome then reads these instructions and translates them into a protein. This translation process involves a number of steps. Proteins called eukaryotic translation initiation factors (or eIFs for short) coordinate the first step in the process, which is known as 'initiation'.

The eIFs also provide the cell with ways to control how quickly it makes proteins. For example, when a cell is stressed, either by starvation or toxins, it adds a phosphate group onto part of an eIF protein, called eIF2α. This modification makes this eIF protein less able to initiate translation, and so the cell builds fewer proteins and conserves more of its resources during times of stress.

Once the stressful conditions are over, the phosphate group is removed from eIF2α by an enzyme called a phosphatase. This phosphatase contains two subunits: one that recognizes eIF2α and another that removes the phosphate group. However, experiments that attempted to recreate this phosphatase activity using just these two subunits in a test tube failed to generate a working enzyme that specifically targeted the phosphate group of eIF2α. This suggests that in cells this enzyme contains an additional unknown subunit. Now, Chambers, Dalton et al. (and Chen et al.) report the identity of a 'missing' third subunit as a protein known as globular-actin or G-actin.

Chambers, Dalton et al. engineered human and fruit fly cells to add 'molecular handles' on the two known subunits of the phosphatase enzyme. These handles could then be used to essentially pull these proteins out of the mixture of molecules within a cell and see what other proteins came along too. Both of the known subunits 'pulled' G-actin along with them; this suggested that it could be the missing part of the phosphatase enzyme.

Further experiments confirmed that G-actin works together with the other two subunits to specifically remove the phosphate group from eIF2α in mouse cells that had been stressed using a harmful chemical. Individual G-actin proteins can bind together to form long filaments, and signals that encourage a cell to divide or move also trigger the formation of actin filaments. This reduces the activity of the phosphatase enzyme by depriving it of a crucial component, i.e., free G-actin proteins. As such, the new mechanism described by Chambers, Dalton et al. suggests how growth and movement signals might also change a cell's sensitivity to stress. These findings may hopefully enable stressed cells to be targeted by drugs to treat disease; but future work is needed to clarify under what circumstances the integration of such signals into the stress response is beneficial to the cell.

*Novoa et al., 2001*; *Jousse et al., 2003*). In *Drosophila*, a single PPP1R15 has been described that is required for anabolic larval growth (*Malzer et al., 2013*), while in mammals, two PPP1R15 paralogues exist: a constitutively expressed isoform PPP1R15B (also known as CReP) and a stress-inducible isoform PPP1R15A (also GADD34) (*Novoa et al., 2001*; *Jousse et al., 2003*). PPP1R15 family members share significant homology in their C-terminal conserved PP1-interacting domain, constituting a core functional domain sufficient to dephosphorylate eIF2α when over expressed in cells (*Novoa et al., 2001*; *Malzer et al., 2013*). In contrast, the less well-conserved N-terminal portion of each PPP1R15 determines protein stability (*Brush and Shenolikar, 2008*) and subcellular localisation (*Zhou et al., 2011*), although the importance of these functions in the regulation of eIF2α phosphatase activity within the cell remains to be worked out.

The importance of eIF2α dephosphorylation is highlighted by PPP1R15 loss-of-function phenotypes. In *Drosophila*, ubiquitous RNAi-mediated depletion of dPPP1R15 leads to embryonic lethality, while failure of blastocyst implantation is seen in *Ppp1r15a-Ppp1r15b* double knockout mouse embryos (*Harding et al., 2009*; *Malzer et al., 2013*). Deficiency of PPP1R15B in isolation permits survival to gestation but leads to defects of haematopoiesis and death in the early neonatal period (*Harding et al., 2009*). In contrast, PPP1R15A-deficient mice are overtly healthy when raised in standard laboratory conditions and show increased resistance to ER stress-induced tissue damage (*Marciniak et al., 2004*).

PPP1R15A is regulated transcriptionally (*Novoa et al., 2001*), but relatively little is known about post-transcriptional regulation of its activity or the regulation of the constitutively expressed

PPP1R15B or *Drosophila* dPPP1R15 (*Jousse et al., 2003*; *Malzer et al., 2013*). The literature offers numerous examples of proteins that associate with one or other of the PPP1R15 family members (*Hasegawa et al., 2000a*, *2000b*; *Wu et al., 2002*; *Hung et al., 2003*; *Shi et al., 2004*), but these are largely single studies with no follow-up or physiological validation. In this study, we set out to characterise conserved elements of the PPP1R15 interactome and in doing so identified a novel mechanism for the regulation of eIF2α phosphatases that links the ISR with cytoskeletal dynamics.

## Results

### PPP1R15 selectively associates with monomeric G-actin in cells

Important regulators/components of the PPP1R15-PP1 holoenzyme are likely to be conserved between species and paralogues; therefore, we set out to identify proteins that interact with both mammalian paralogues, PPP1R15A and PPP1R15B, and their non-vertebrate homologue, *Drosophila* dPPP1R15. GFP-tagged human PPP1R15A and PPP1R15B were expressed in human embryonic kidney (HEK) 293T cells and subjected to GFP-Trap affinity purification followed by mass spectrometry (*Figure 1A,B* and *Figure 1—figure supplements 1, 2*), whereas V5-His-tagged dPPP1R15 was expressed in *Drosophila* Schneider 2 (S2) cells and subjected to affinity purification using anti-V5-His resin followed by mass spectrometry (*Figure 1A*). In addition to the anticipated association of PP1, we identified a number of other proteins that were bound to each PPP1R15 bait (as defined by >twofold enrichment over control and the detection of ≥5 identifiable peptides in the mass spectra; *Figure 1—figure supplements 1, 2*).

Actin emerged as the prominent partner conserved across phyla (*Figure 1A,B*). Confidence in this association was bolstered by finding that *Drosophila* dPPP1R15 also associated with mammalian actin in stoichiometric amounts (*Figure 1C*). This association was observed regardless of which terminus of dPPP1R15 was tagged. Actin's presence in complex with PPP1R15 was also observed using other tag combinations: an N-terminal fusion of GST with the catalytic subunit PP1A expressed in HEK293T cells alongside PPP1R15A yielded a complex containing GST-PP1A, PPP1R15A, and actin upon glutathione-affinity chromatography (*Figure 1D*).

GFP-tagged PPP1R15A purified from HEK293T cells failed to associate with filamentous F-actin in a co-sedimentation assay (*Figure 2A*) suggesting selective interaction between PPP1R15 and monomers of soluble G-actin. The distribution of actin between its monomeric G or polymeric F form is influenced by physiological conditions and can be biased pharmacologically by small molecules that stabilise either form (*White et al., 1983*). Jasplakinolide, which stabilises F-actin filaments and depletes the cells of G-actin (*Holzinger, 2009*), abolished the interaction between PPP1R15A and actin (*Figure 2B*, lane 4). In contrast, latrunculin B, which binds to the nucleotide-binding cleft of actin, thus increasing the cytoplasmic pool of G-actin (*Nair et al., 2008*), potently enhanced the recovery of actin in complex with PPP1R15A (*Figure 2B*, lane 3). Cytochalasin D also increases the cellular pool of G-actin, but does so by engaging actin's barbed end, competing with several known G-actin-binding proteins (*Miralles et al., 2003*; *Dominguez and Holmes, 2011*; *Shoji et al., 2012*); exposure to cytochalasin diminished the recovery of actin in complex with PPP1R15A (*Figure 2B* lane 2).

Actin polymerisation is sensitive to physiological growth cues (*Sotiropoulos et al., 1999*). Serum starvation, which resulted in the anticipated conversion of F to G-actin (*Figure 2C*) enhanced recovery of actin in complex with PPP1R15A in NIH-3T3 cell lysates (*Figure 2D*). On serum re-feeding, cables of F-actin re-formed within the cytoplasm and less actin was recovered in complex with PPP1R15A. The aforementioned confirm that both mammalian and insect PPP1R15 regulatory subunits engage G-actin and that the interaction between them is sensitive to physiological changes in the availability of G-actin.

### Actin associates with the conserved C-terminal functional core of PPP1R15

Human PPP1R15A is a 674 amino acid protein, comprising an N-terminal domain required for membrane interaction, a region of proline, glutamate, serine, threonine (PEST) rich repeats of uncertain function, and a C-terminal functional core domain that interacts with the PP1 catalytic subunit (*Figure 3A*) and is sufficient for mediating substrate-specific dephosphorylation (*Novoa et al., 2001*; *Kojima et al., 2003*; *Ma and Hendershot, 2003*). Deletion analysis showed that the

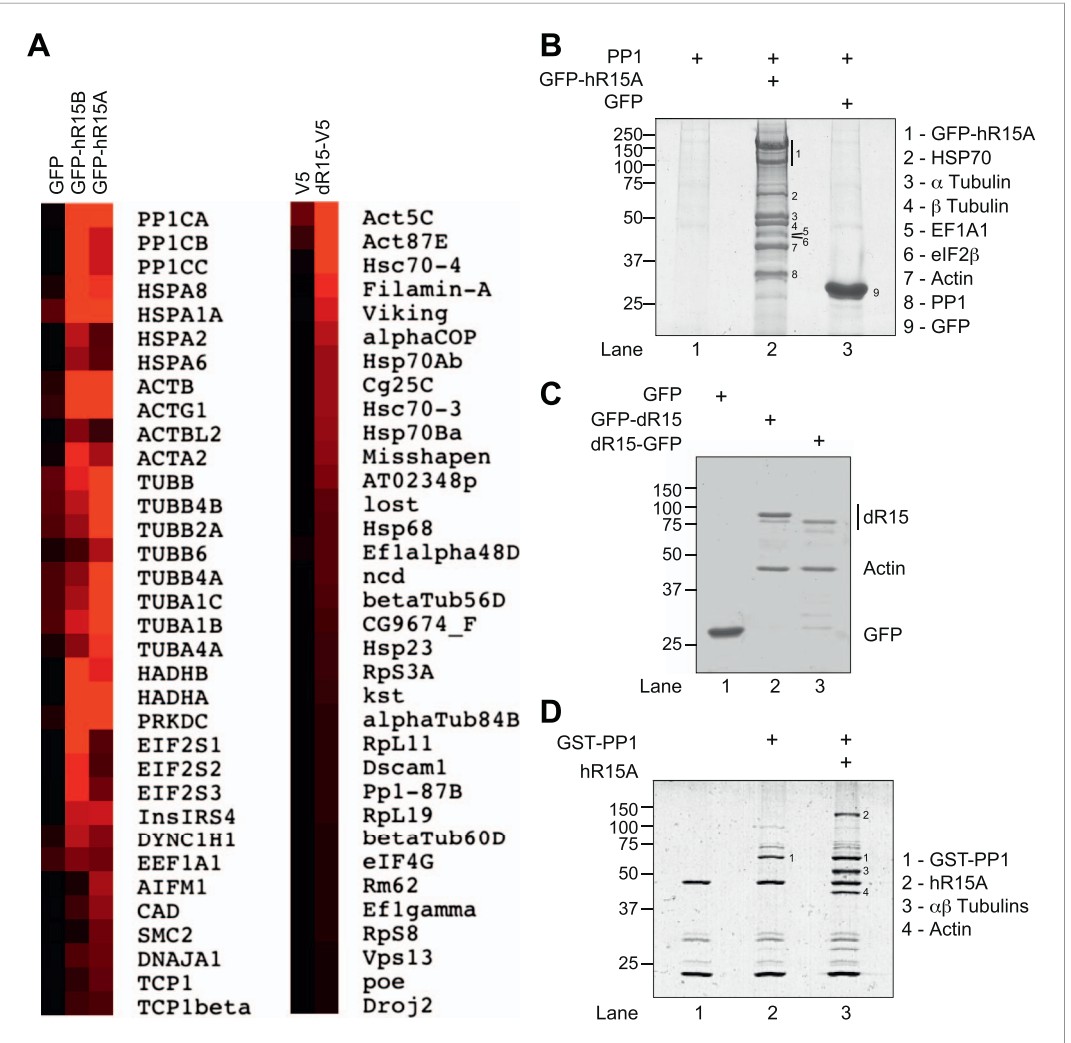

**Figure 1**. PPP1R15 associates with actin in mammalian and insect cells. (**A**) Heat map of proteins associated with GFP, GFP-tagged human PPP1R15B (GFP-hR15B) and GFP-tagged human PPP1R15A (GFP-hR15A) affinity-purified from transiently transfected HEK293T cells (left panels); heat map of proteins associated with V5 and V5-tagged *Drosophila* PPP1R15 (dR15-V5) affinity purified from transiently transfected S2 cells (right panels). Samples were analysed by Orbitrap mass spectrometer. Intensity reflects total spectrum count of identified peptides. Proteins identified by at least five spectra and showing at least twofold enrichment over control are shown. (**B**) Coomassie-stained SDS-PAGE of GFP-affinity purified proteins from HEK293T cells expressing indicated proteins. Indicated bands were individually excised and identified by mass spectrometry. (**C**) Coomassie-stained SDS-PAGE of GFP-affinity purified proteins from HEK293T cells expressing indicated proteins. Bands were individually excised and identified by mass spectrometry. (**D**) Coomassie-stained SDS-PAGE of glutathione-affinity purified proteins from HEK293T cells. Indicated bands were individually excised and identified by mass spectrometry.

The following figure supplements are available for figure 1:

**Figure supplement 1**. Mass spectrometry results of GFP, GFP-PPP1R15B, and GFP-PPP1R15A expressed in HEK293T cells and purified using GFP-Trap beads.

**Figure supplement 2**. Mass spectrometry results of V5 and dPPP1R15A-V5 expressed in S2 cells and purified using anti-V5 immunoprecipitation.

---

C-terminus of PPP1R15A (residues 501–674) was also sufficient for the association with actin (*Figure 3B*). Further deletion revealed that residues C-terminal to amino acid 615 were essential for actin association but not for PP1 binding, which was enfeebled but not abolished (*Figure 3B,C*).

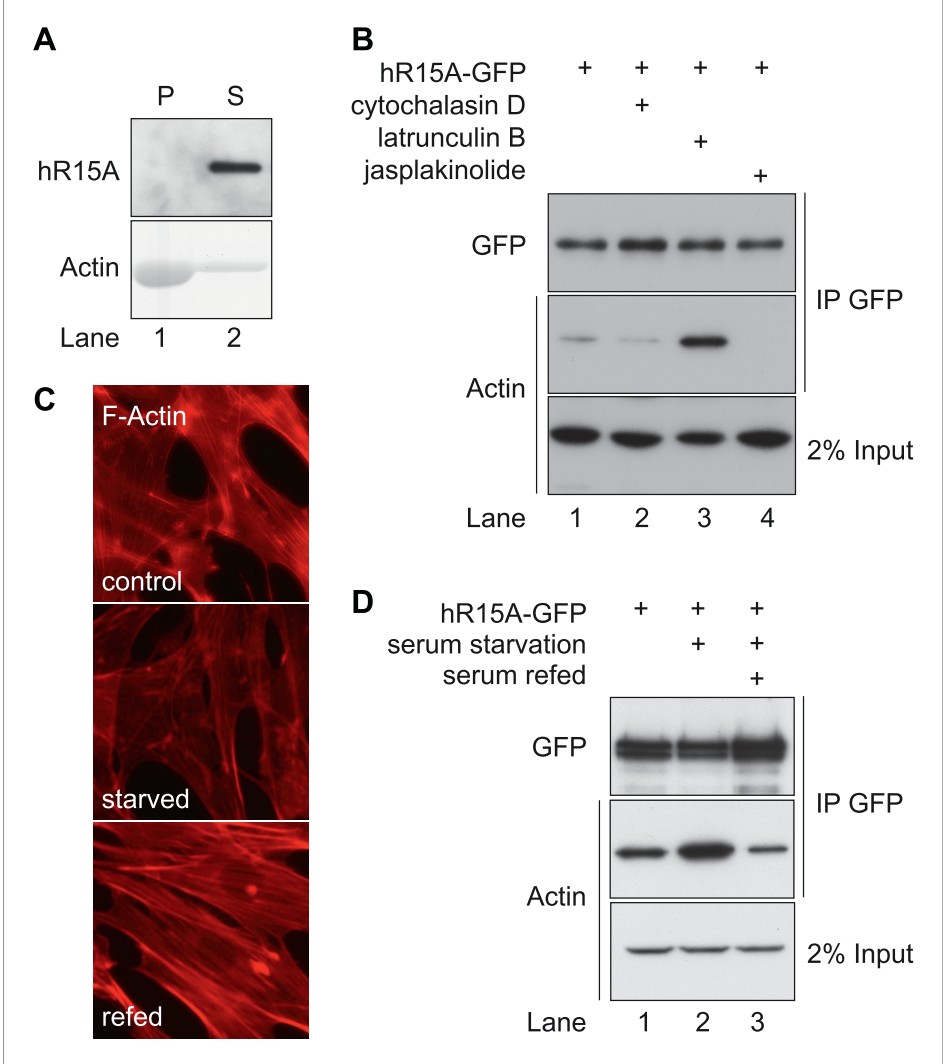

**Figure 2.** PPP1R15 selectively associates with monomeric G-actin in cells. (**A**) Immunoblot (upper panel) and Coomassie-stained gel (lower panel) of affinity-purified GFP-tagged PPP1R15A and purified actin. Samples were incubated and centrifuged to pellet F-actin (lane 1), leaving G-actin in the supernatant (lane 2); pellet P, supernatant S. (**B**) Immunoblot for GFP and actin of GFP-affinity purified proteins (upper two panels) from HEK293T cells expressing GFP-tagged PPP1R15A (hR15A-GFP) treated with 2 µM of each indicated compound. Immunoblot for actin of 2% of input. (**C**) Fluorescence microscopy image of NIH-3T3 cell F-actin arrangement. NIH-3T3 cells were left untreated (control), cultured in serum-free medium for 24 hr (serum starved), cultured in serum-free medium for 18 hr, followed by addition of medium containing 10% vol/vol FBS for 6 hr (serum refed), then fixed and stained with Alexa-Fluor 568 phalloidin and imaged by confocal microscopy. (**D**) Immunoblot for GFP and actin of NIH-3T3 lysates from cells treated as in 'C' then subjected to GFP affinity purification (upper two panels). Immunoblot for actin of 2% of input.

Incorporation of five residues ($W^{616}$–$R^{620}$ of human PPP1R15A) restored fully the recovery of actin in complex with PPP1R15A (*Figure 3C* lane 6), while the $W^{616}A$ and $L^{619}A$ double mutation strongly enfeebled actin recovery in complex with PPP1R15A (*Figure 3D*). A $V^{556}E$ mutation of the RVxF motif, which all but abolishes PP1 binding and eIF2α dephosphorylation in vivo (*Novoa et al., 2001*), also attenuated recovery of actin in complex with PPP1R15A, but failed to abolish it altogether (*Figure 3C*, lane 3).

The quantities of actin and PP1 recovered in complex with PPP1R15A were sensitive to the salt concentration of the buffers used (*Figure 3—figure supplement 1*). Actin association with PPP1R15A dropped progressively with increasing salt (75% of the actin bound at 150 mM salt was

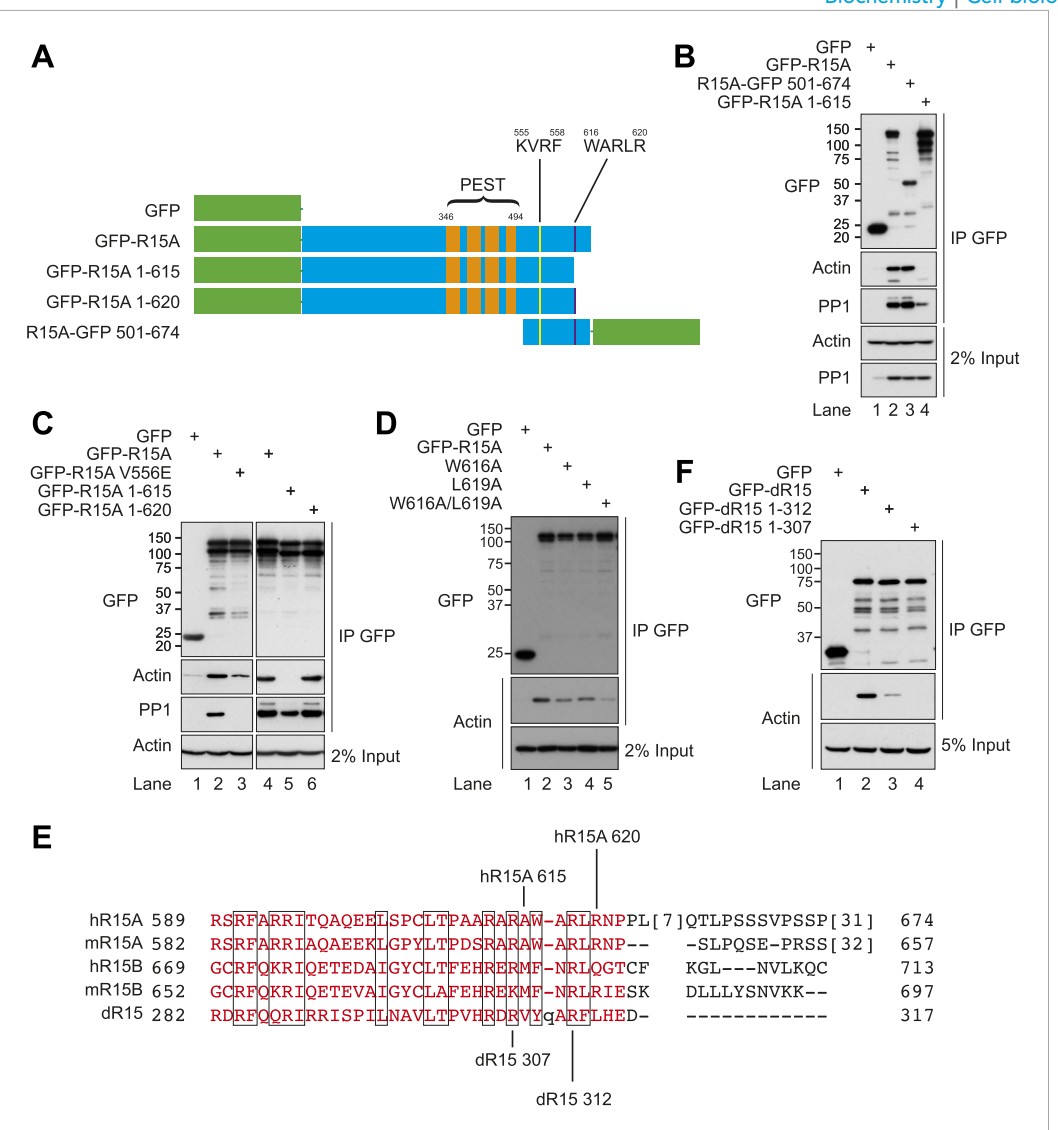

**Figure 3**. Actin associates with the conserved C-terminal portion of PPP1R15. (**A**) Schematic diagram of human PPP1R15A (R15A) constructs used. Green indicates GFP. PEST repeats (between residues 346 and 494, orange), K$^{555}$VRF$^{558}$ (yellow), and W$^{616}$ARLR$^{620}$ (purple) sequences are identified. (**B**) Immunoblot for GFP, actin, and PP1 of HEK293T lysates from cells expressing indicated constructs and PP1, and subjected to GFP affinity purification (upper three panels). Immunoblot for actin and PP1 of 2% of input. (**C**) Immunoblot for GFP, actin, and PP1 of HEK293T lysates from cells expressing indicated constructs and PP1, and subjected to GFP affinity purification (upper three panels). Immunoblot for actin of 2% of input. (**D**) Immunoblot for GFP and actin of HEK293T lysates from cells expressing indicated constructs and subjected to GFP affinity purification (upper two panels). Immunoblot for actin of 5% of input (lower panel). (**E**) Sequence alignment of C-terminal portions of human (h) and murine PPP1R15A (mR15A) and PPP1R15B (mR15B) and *Drosophila* dPPP1R15 (dR15) with regions of homology boxed. Specific truncations are indicated. (**F**) Immunoblot for GFP and actin of HEK293T lysates from cells expressing indicated constructs and subjected to GFP affinity purification (upper two panels). Immunoblot for actin and PP1 of 2% of input.

The following figure supplements are available for figure 3:

**Figure supplement 1**. Immunoblot for GFP, actin, and PP1 of GFP-Trap pull-downs and 2% of input.

**Figure supplement 2**. Immunoblot for GFP, actin, and PP1 of GFP-Trap pull-downs and 2% of input.

lost at 350 mM), as did PP1 association, with no detectable binding at 350 mM. The complex was stable in non-denaturing detergents (triton X-100 and digitonin), but washes in a buffer containing the harsher detergents, sodium deoxycholate (0.5% vol/vol) and SDS (0.1% vol/vol), completely abolished interaction between PPP1R15A and both PP1 and actin (*Figure 3—figure supplement 2*).

*Drosophila* dPPP1R15 is half the size of the mammalian PPP1R15s. When aligned, mammalian PPP1R15A, PPP1R15B, and dPPP1R15 share significant homology within their C-termini, which drops off at residue 622 of human PPP1R15A (*Figure 3E*). We therefore truncated the *Drosophila* protein within and immediately N-terminal to this region of homology ($Y^{307}$–$H^{312}$). Partial truncations reduced the association of dPPP1R15 with actin, while deletion of the entire segment (at residue 307) completely abolished the interaction (*Figure 3F*). The interaction with actin, thus maps to the conserved portion of PPP1R15 family members and is favoured by a short stretch of hydrophobic residues at the extreme C-terminus of this core. Mutational analysis thus points to a measure of independent association of PP1 or actin with PPP1R15, but highlights the enhanced recovery of the three proteins in a ternary complex of PPP1R15, PP1, and actin.

## Association of G-actin with PPP1R15 regulates eIF2α phosphatase activity in vivo

To examine the relevance of G-actin to the endogenous PPP1R15 complex, wild-type *Ppp1r15a$^{+/+}$* and mutant *Ppp1r15a$^{mut/mut}$* mouse embryonic fibroblasts (MEFs) were treated with the ER stress promoting agent tunicamycin to induce the ISR and expression of PPP1R15A. The *Ppp1r15a$^{mut/mut}$* cells express a C-terminal truncated PPP1R15A that is incapable of binding PP1 (*Novoa et al., 2003*) and served as a negative control. As expected, a robust PP1 signal was found associated with endogenous wild-type PPP1R15A in the stressed cells, whilst no signal was detected in PPP1R15A immunoprecipitates from the *Ppp1r15a$^{mut/mut}$* cells (*Figure 4A*, lanes 2 and 5). The poor reactivity of the available antisera to actin and tendency of actin to associate non-specifically with immunoprecipitation reactions frustrated our efforts to detect actin associated with endogenous PPP1R15A in MEFs; however, treatment with jasplakinolide, which depleted the soluble pool of actin led to a marked loss of PP1 association with PPP1R15A in the stressed cells (compare lanes 2 and 3, *Figure 4A*). To test the converse interaction, PP1 was affinity purified from MEF lysates using microcystin–agarose beads. Whilst the presence of other known PP1-actin complexes precludes meaningful interpretation of actin purified by microcystin affinity (*Oliver et al., 2002*; *Kao et al., 2007*), the PPP1R15A-PP1 interaction detected in stressed wild-type cells was attenuated by jasplakinolide-driven depletion of soluble actin (*Figure 4B*). Actin's role in the stability of the PPP1R15A-PP1 complex was confirmed in HEK293T cells (*Figure 4C*).

In order to address the association of actin with endogenous PPP1R15A directly, we used HEK293T cells, which generated less background actin signal in control immunoprecipitation reactions. Purified GFP-tagged PPP1R15 was used as a standard to determine the minimum amount of PPP1R15 that permitted detection of associated actin (*Figure 4D*). Scaling of input material to immunopurify similar quantities of endogenous and overexpressed PPP1R15A led to recovery of similar amounts of associated endogenous actin (*Figure 4D*). This supports a role for the interaction in cell physiology.

A functional role for actin in PPP1R15 complexes was suggested by the observation that depletion of cellular G-actin by exposure to jasplakinolide promoted a rapid increase in the levels of phosphorylated eIF2α (*Figure 5A,B*). To extend these observations, cells were treated with the SERCA pump inhibitor thapsigargin, which depletes the ER of calcium and rapidly and transiently activates the ER stress-inducible kinase PERK. As expected, this led to a robust yet transient phosphorylation of eIF2α by PERK (*Figure 5C* lanes 1–6). The transient nature of this phosphorylation relates to the rectifying response of PERK on levels of ER stress, but also draws on the combined activities of constitutively expressed PPP1R15B and the induction of PPP1R15A that promote eIF2α dephosphorylation (*Novoa et al., 2001*; *Jousse et al., 2003*; *Novoa et al., 2003*). In the presence of jasplakinolide, the elevated levels of phosphorylated eIF2α induced by thapsigargin persisted (*Figure 5C*, lanes 7–12), while latrunculin B had no effect on the time course of eIF2α phosphorylation (*Figure 5—figure supplement 1*). It is noteworthy that peak levels of eIF2α phosphorylation were higher in cells treated with jasplakinolide (compare lanes 1–2 with 8–9 of *Figure 5C*). This occurred well before the induction of PPP1R15A suggesting that either an endogenous basally expressed phosphatase or the kinase was affected.

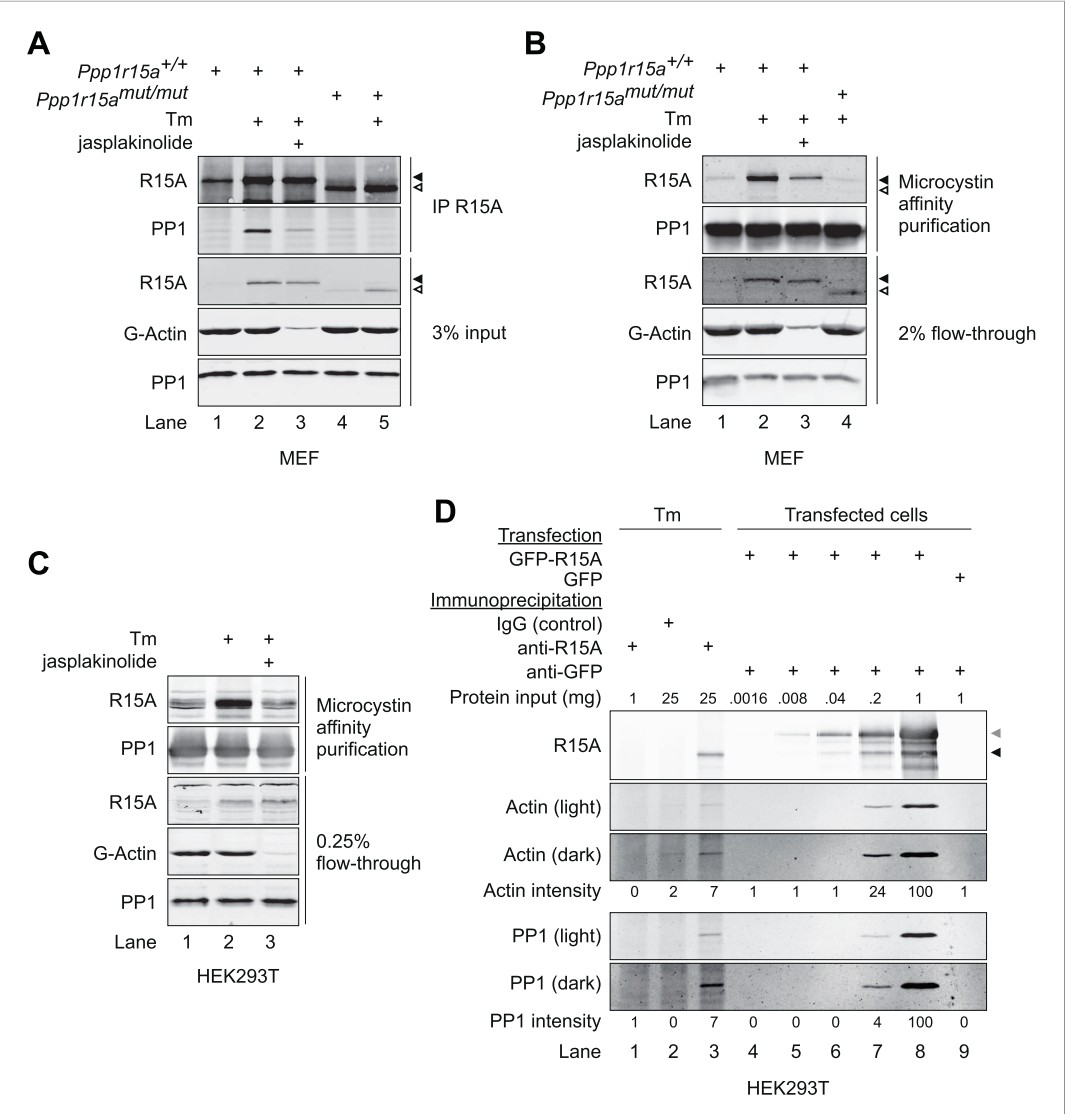

**Figure 4**. G-actin stabilises the PPP1R15A-PP1 complex in vivo. (**A**) Immunoblots of endogenous PPP1R15A (R15A) and associated PP1 immunopurified from wild type (*Ppp1r15a*^+/+) or mutant mouse embryonic fibroblasts homozygous for a C-terminal truncation of PPP1R15A that abolishes interaction with PP1 (*Ppp1r15a*^mut/mut) with an anti-PPP1R15A antiserum (IP R15A). Where indicated, cells were treated with tunicamycin 2 μg/ml (Tm) for 8 hr to induce PPP1R15A and jasplakinolide (1 μM) for 1.5 hr before harvest. The lower three panels are immunoblots of the input of the immunoprecipitation reactions analysed in the top two panels. Closed and open triangles mark, respectively, the wild type and mutant PPP1R15A lacking the C-terminal functional core. To assess G-actin content of the input, the sample was subjected to ultracentrifugation to remove F-actin. (**B**) PPP1R15A and PP1 immunoblots of PP1-containing complexes purified by microcystin affinity chromatography from cells as in '**A**' above. The lower three panels report on the content of input material. (**C**) As in '**B**', above, but reporting on PP1-continaing complexes purified by microcystin affinity chromatography from HEK293T cells. (**D**) Immunoblots of endogenous or over-expressed GFP-tagged PPP1R15A and associated endogenous PP1 and actin immunopurified with antiserum to PPP1R15A, non-immune rabbit IgG (as a control) or antiserum to GFP from lysates of tunicamycin-treated HEK293T cells (Tm, 2.5 μg/ml for 8 hr to induce endogenous PPP1R15A) or cells transfected with plasmids expressing GFP-PPP1R15A (GFP-R15A) or GFP. The protein content of the cell lysate applied to the immunoprecipitations is noted above the immunoblots ('Protein input'). Endogenous PPP1R15A and the larger GFP-PPP1R15A are marked by black and grey arrowheads, respectively. Both heavy and light exposures of the actin and PP1 immunoblots are provided and the relative intensity of the signals is noted.

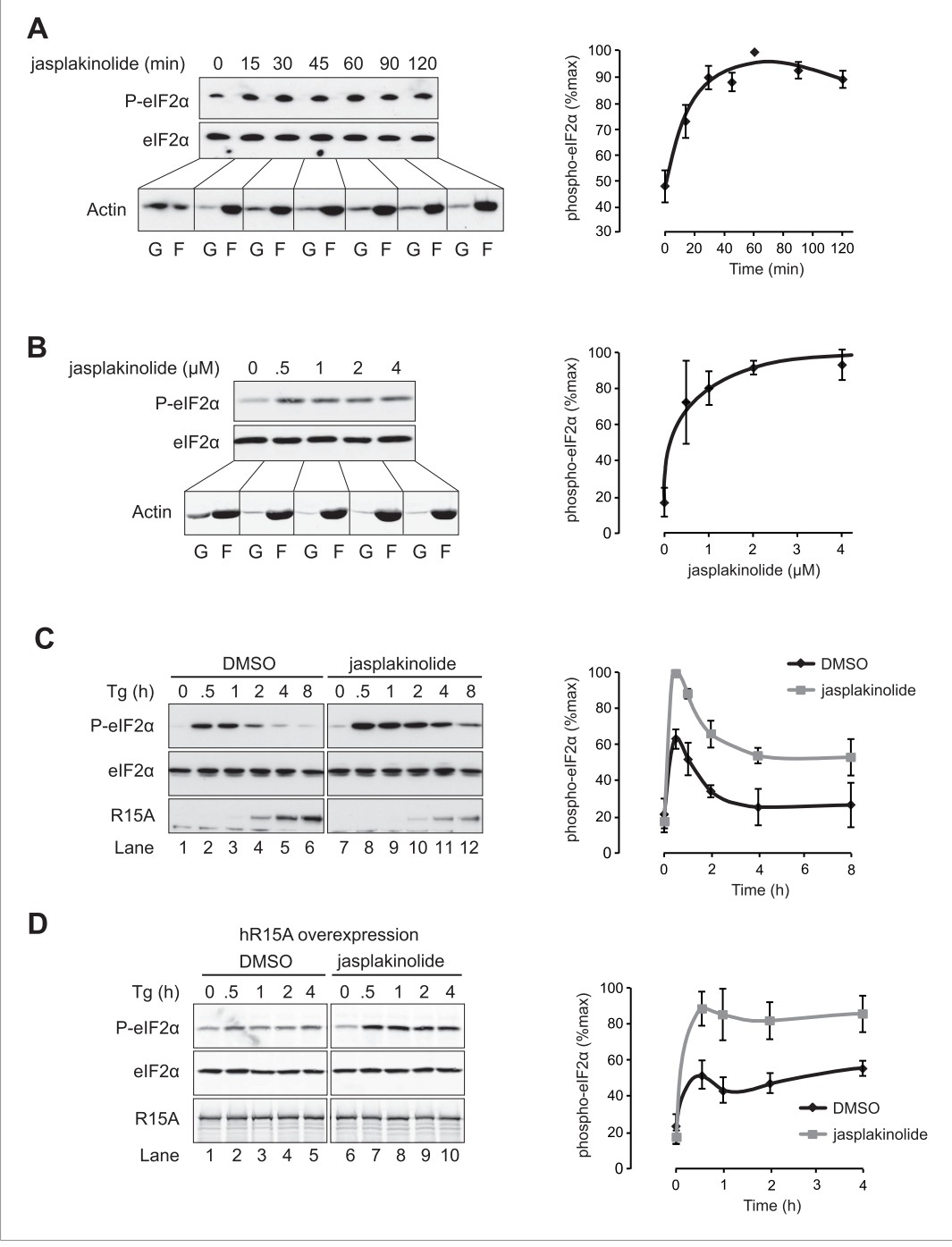

**Figure 5**. Association of G-actin with PPP1R15 regulates eIF2α phosphatase activity. (**A**) Immunoblot for phosphorylated eIF2α (P-eIF2α), total eIF2α, and actin. Wild-type (WT) mouse embryonic fibroblasts (MEF) were treated with jasplakinolide 1 µM for the indicated times. Lysates were subjected to sedimentation assay and immunoblot for G-actin in the supernatant (G) or F-actin in the pellet (F). (**B**) Immunoblot for phosphorylated eIF2α (P-eIF2α), total eIF2α, and actin. WT MEFs were treated with the indicated concentrations of jasplakinolide for 1 hr. Lysates were analysed as in '**A**'. (**C**) Immunoblot for phosphorylated eIF2α (P-eIF2α), total eIF2α, and PPP1R15A. WT MEFs were treated with thapsigargin 400 nM for the indicated times, without or with jasplakinolide 1 µM. (**D**) Immunoblot for P-eIF2α and PPP1R15A (hR15A). GFP-hPPP1R15A Tet-On HeLa cells were treated with doxycycline to induce transgene

*Figure 5. continued on next page*

*Figure 5. Continued*

expression and then with thapsigargin 400 nM for the indicated times. Cells were co-treated with jasplakinolide or latrunculin B 1 µM or vehicle as indicated. Accompanying graphs show means ±SEM of n = 3 independent repeats.

The following figure supplement is available for figure 5:

**Figure supplement 1**. Immunoblot for phosphorylated eIF2α (P-eIF2α), total eIF2α and PPP1R15A.

Induction of PPP1R15A was observed in thapsigargin-treated cells from 2 hr onwards but was diminished in cells co-treated with jasplakinolide (likely, a consequence of profound attenuation of protein synthesis). To minimise the confounding effect of diminished PPP1R15A levels in jasplakinolide-treated cells at later time points, the regulatory subunit was conditionally over-expressed under the control of a tetracycline-responsive promoter. As expected, enforced expression of PPP1R15A abrogated the phosphorylation of eIF2α in response to thapsigargin; however, jasplakinolide reversed the inhibitory effect of PPP1R15A over-expression restoring elevated levels of phosphorylated eIF2α to thapsigargin-treated cells (*Figure 5D*).

To examine more closely the effects of the actin cytoskeleton on dephosphorylation of eIF2α within living cells, we sought to gain temporal control over the phosphorylation phase of the cycle. To this end, we made use of a small molecule eIF2α kinase inhibitor, GSK2606414A (*Axten et al., 2012*). At the concentration used, this inhibits the eIF2α kinases PERK and PKR, but not GCN2. Thus, application of GSK2606414A to *Gcn2*−/− fibroblasts rapidly abrogates eIF2α phosphorylation. To measure selectively the dephosphorylation phase of the stress response, PERK-mediated phosphorylation of eIF2α was induced by thapsigargin and further phosphorylation was then blocked with the kinase inhibitor GSK2606414A. In the absence of kinase activity, the subsequent decay of the phosphorylated eIF2α signal reflects its de-phosphorylation (degradation of the protein is not observed over this time scale), which was markedly attenuated by jasplakinolide (*Figure 6A*, compare lanes 1–5 with lanes 6–10 and *Figure 6B*).

Jasplakinolide-mediated induction of ATF4 was abrogated in cells in which the serine 51 phosphorylation site of eIF2α had been mutated to alanine (*Figure 6—figure supplement 1*), thus validating ATF4 as an indicator of the effects of manipulation of the actin cytoskeleton on ISR activity. ATF4, whose levels decline rapidly upon GSK2606414A-mediated shutdown of kinase activity, was also stabilised by jasplakinolide (*Figure 6A*, lowest panel), reflecting the functional significance of the defect in eIF2α dephosphorylation imposed by the depletion of G-actin. Levels of phosphorylated eIF2α induced by jasplakinolide were undiminished in cells lacking any one of the four known eIF2α kinases (*Figure 6C*), suggesting that the compound's effects on levels of phosphorylated eIF2α reflect its workings on the dephosphorylation phase of the stress cycle and not to off-pathway stress culminating in kinase activation.

Actin was recovered in complex with both the inducible and constitutive mammalian PPP1R15 family members (*Figures 1 and 7A*). To determine if the effects of G-actin were preferentially mediated by complexes containing one or the other PPP1R15 subunit, we compared the effect of jasplakinolide on levels of phosphorylated eIF2α in wild-type MEFs and MEFs deficient in one or the other regulatory subunit. Enhanced levels of phosphorylated eIF2α in jasplakinolide-treated cells and the synergistic effects of depleting G-actin on the response to thapsigargin were observed in wild-type cells and in cells lacking either PPP1R15A or PPP1R15B-directed eIF2α dephosphorylation (*Figure 7B–D*). These observations indicate that G-actin plays a functional role in holophosphatases constituted with either regulatory subunit.

To explore in further detail the basis for the correlation between G-actin levels and eIF2α dephosphorylation, we compared in vitro eIF2α-directed phosphatase activity of PPP1R15A-containing complexes recovered from untreated and jasplakinolide-treated cells. PPP1R15A-GFP fusion protein was expressed transiently in HEK293T cells overnight. The following day, cells were treated either with vehicle or with 1 µM jasplakinolide for 1 hr, lysed and then subjected to GFP-affinity purification using GFP-Trap beads. The resulting complexes were divided between four tubes and incubated for the indicated times at 37°C with pre-phosphorylated recombinant eIF2α (see 'Materials and methods'). Less actin and PP1 were recovered in complex with tagged PPP1R15A from jasplakinolide-treated cells (whilst HSP70 binding was unaffected) (*Figure 8A*), and the eIF2α-directed phosphatase activity of the purified complexes was likewise diminished

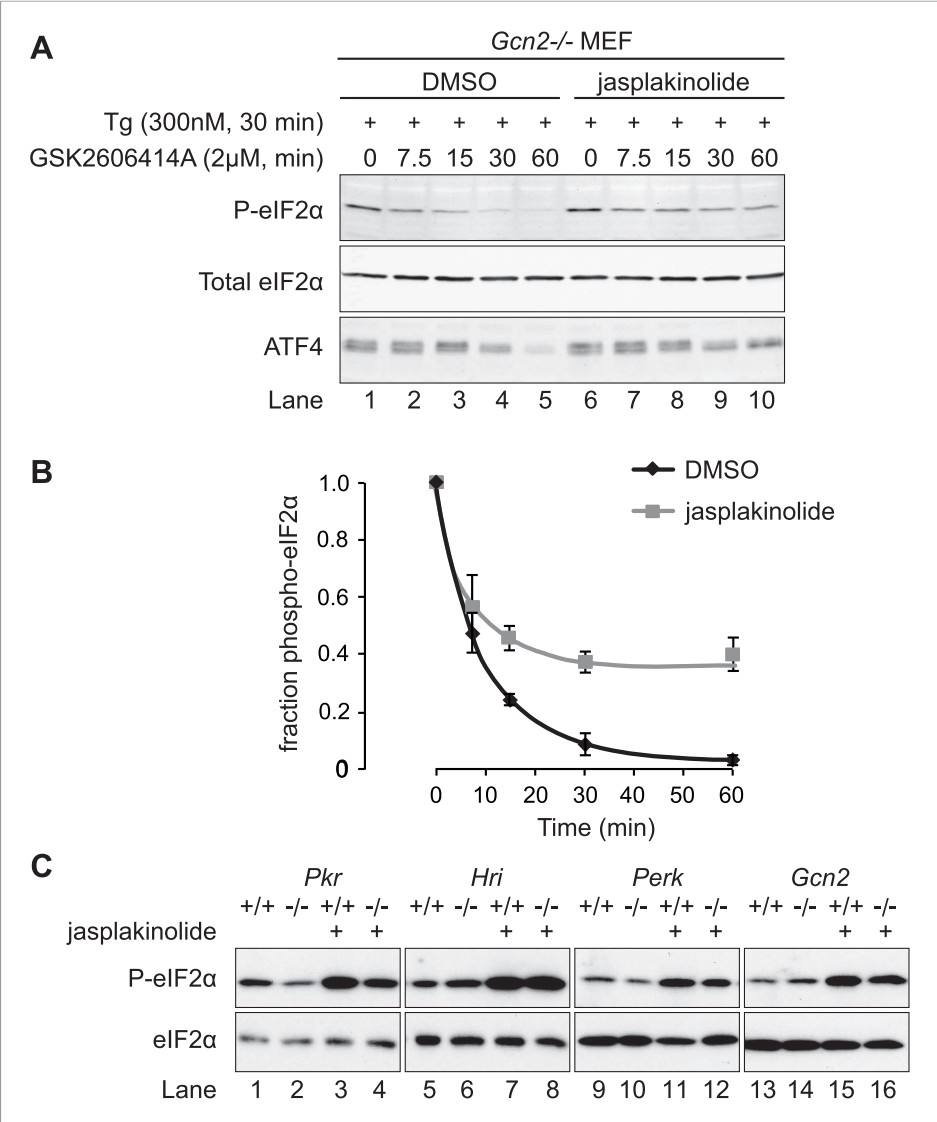

**Figure 6**. Jasplakinolide diminishes eIF2α phosphatase activity in vivo. (**A**) Immunoblot for phosphorylated eIF2α (P-eIF2α), total eIF2α, and ATF4. *Gcn2–/–* MEFs were pre-treated with thapsigargin 300 nM for 30 min to induce eIF2α phosphorylation and ATF4 protein levels. GSK2606414A at 2 μM was then added for the indicated times. Protein lysates were analysed by SDS-PAGE and subjected to immunoblot. (**B**) Quantification of '**A**' using ImageJ software. Mean ± SEM of n = 3 independent repeats. (**C**) Immunoblot for phosphorylated eIF2α (P-eIF2α) and total eIF2α. MEFs of the indicated genotypes were treated with or without jasplakinolide 1 μM for 1 hr. Protein lysates were analysed by SDS-PAGE and subjected to immunoblot.

The following figure supplement is available for figure 6:

**Figure supplement 1**. Immunoblot for P-eIF2α, total eIF2α, and ATF4 (specific band marked with an asterisk) in lysates of wild type (WT) or eIF2αAA MEFs following treatment with thapsigargin 300 nM for 4 hr and/or jasplakinolide 1 μM for 4 hr.

(*Figure 8A,B*). Complex formation with PP1 contributes to dPPP1R15 stability; however, the decline in PPP1R15A levels in cells exposed to the translational inhibitor cycloheximide, was unaffected by the presence of jasplakinolide (*Figure 8C,D*) indicating that stabilisation of a complex between PPP1R15 and PP1 dominates G-actin's role in this experimental system.

## Localised changes in the polymeric status of actin modulate the ISR

In the experiments described thus far, actin polymerisation was manipulated throughout the cell, whereas in vivo the actin cytoskeleton is subject to highly localised changes. In the context of eIF2α

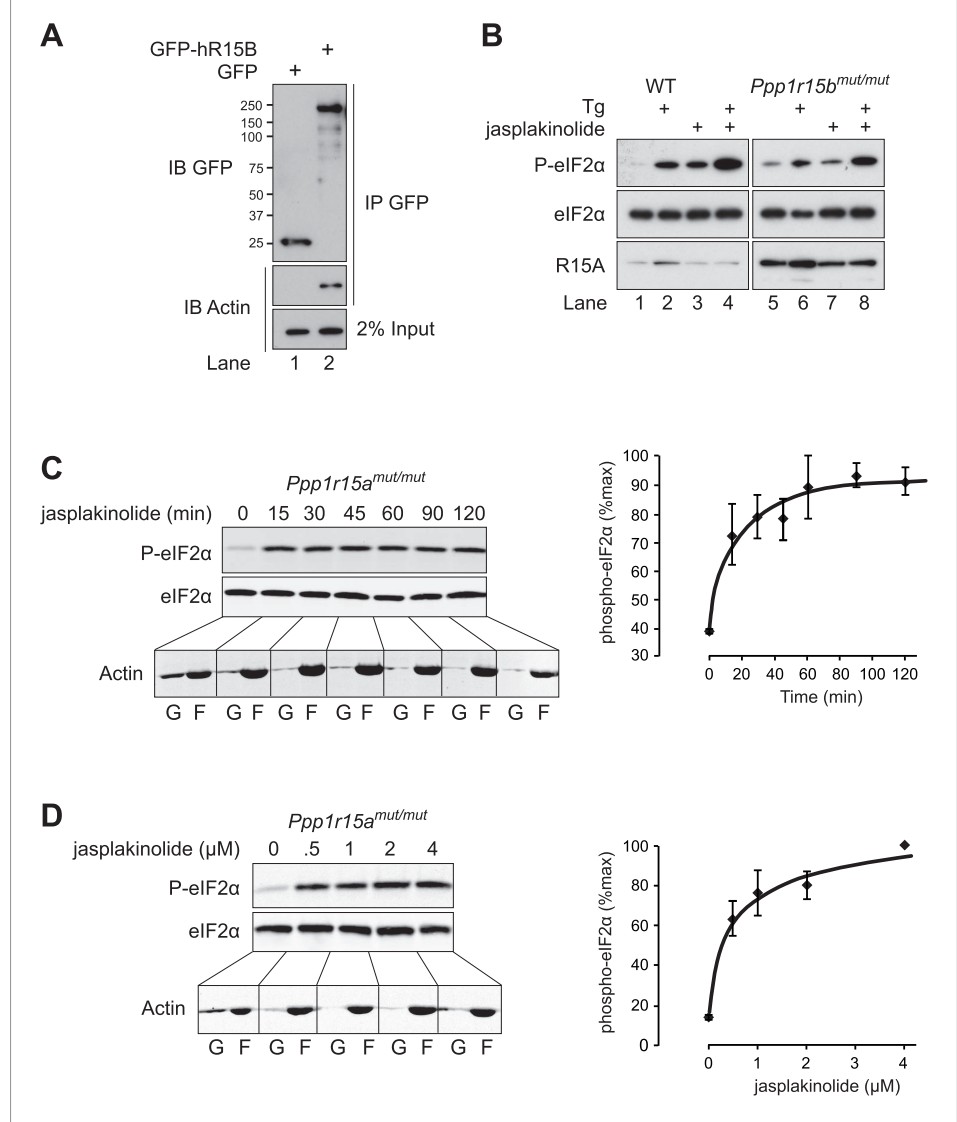

**Figure 7**. Actin associates with PPP1R15B to alter the level of eIF2α phosphorylation. (**A**) Immunoblot for GFP and actin of HEK293T cell lysates expressing either GFP or GFP-PPP1R15B. Upper two panels indicate proteins immunoprecipitated by anti-GFP beads. Lower panel shows 2% of input lysate. (**B**) Immunoblot for P-eIF2α, total eIF2α, and PPP1R15A of lysates from WT or *Ppp1r15b*[tm1Dron/tm1Dron] MEFs treated for 1 hr with thapsigargin 400 nM, jasplakinolide 1 μM or both. (**C**) Immunoblot for phosphorylated eIF2α (P-eIF2α), total eIF2α and actin. *Ppp1r15a*[tm1Dron/tm1Dron] MEFs were treated with jasplakinolide 1 μM for the indicated times. Lysates were subjected to sedimentation assay and immunoblot for G-actin in the supernatant (G) or F-actin in the pellet (F). (**D**) Immunoblot for phosphorylated eIF2α (P-eIF2α), total eIF2α, and actin. *Ppp1r15a*[tm1Dron/tm1Dron] MEFs were treated with the indicated concentrations of jasplakinolide for 1 hr. Lysates were analysed as in '**C**'. Accompanying graphs show mean ± SEM of n = 3 independent repeats.

dephosphorylation, it seemed particularly relevant to examine the effects of actin polymerisation in the vicinity of the ER membrane, where the majority of PPP1R15 is located (*Brush et al., 2003*; *Zhou et al., 2011*; *Malzer et al., 2013*). Cells that conditionally expressed a constitutively active mutant of mDia2, a formin that stimulates localised polymerisation of F-actin (*Pellegrin and Mellor, 2005*), were generated. To direct mDia2 to the same membranous compartments as PPP1R15, we fused the membrane-targeting domain of PPP1R15B (residues 1–146, devoid of catalytic activity) to mDia2 and a GFP tag to facilitate visualisation of the fusion protein. Control cells were generated expressing a PPP1R15B (1–146)-GFP fusion protein lacking mDia2. Induction of eGFP-PPP1R15B

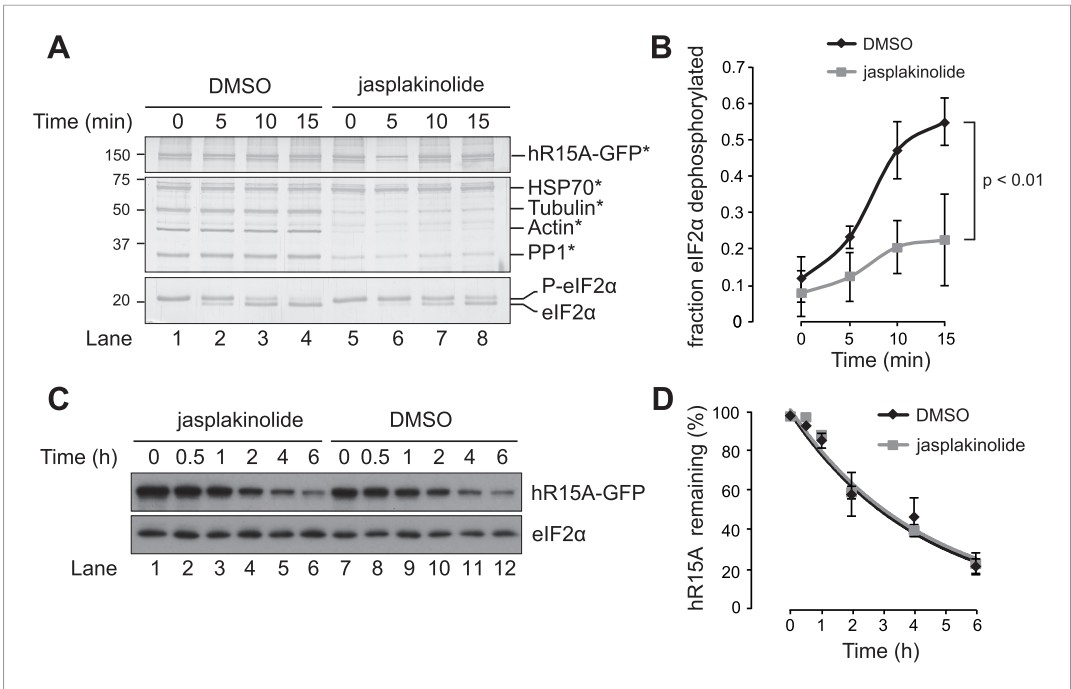

**Figure 8**. Association of actin with PPP1R15A promotes eIF2α phosphatase activity in vitro. (**A**) Silver-stained SDS-PAGE or GFP-affinity purified PPP1R15A-GFP (hR15-GFP) and associated proteins. Asterisks signify identity confirmed by mass spectrometry. Purified complex incubated with phosphorylated recombinant eIF2α N-terminal lobe (eIF2α) for incubated time. Note size shift corresponds to dephosphorylation. (**B**) Quantification of '**A**' using ImageJ software. Mean ± SEM. p value calculated by two-way ANOVA, n = 3. (**C**) Immunoblot for PPP1R15A and eIF2α of HEK293T cells expressing hPPP1R15A-GFP (hR15A-GFP). Treated with cycloheximide 50 μM for indicated times. (**D**) Quantification of '**C**'. Mean ± SEM, n = 3.

(1–146)-mDia2 increased polymerisation of actin adjacent to the ER, as indicated by co-localisation of GFP fluorescence with phalloidin staining (*Figure 9A*). In control cells, the GFP and phalloidin signals showed little correlation, but as expected, these became strongly correlated (indicating co-localisation) on expression of the mDia2 construct (*Figure 9—figure supplement 1*). Immunoblot confirmed the expression of each fusion protein on treatment with doxycycline, but only the mDia2 construct led to the induction of ATF4 (*Figure 9B* compare lanes 4–6 with 13–15). ISRIB, a small molecule that renders cells unresponsive to eIF2α phosphorylation (*Sidrauski et al., 2013*), blocked formin-mediated induction of ATF4 (*Figure 9B*, compare lanes 15 and 16), validating ATF4 as an ISR marker in this assay.

Actin polymerisation in the vicinity of the ER also altered the dynamics of the ISR in response to gradually accruing ER stress induced by the glycosylation inhibitor tunicamycin. During ER stress, phosphorylation of eIF2α by PERK attenuates protein translation to offload the ER (*Harding et al., 1999*). The degree of translational attenuation depends upon the intensity and the rapidity of ER stress (*Novoa et al., 2001*). Sudden and intense ER stress caused by depletion of ER calcium stores by thapsigargin induces marked inhibition of translation. In contrast, gradually escalating ER stress by the accumulation of unglycosylated proteins upon treatment with tunicamycin, attenuates translation less dramatically because induction of PPP1R15A limits the degree of eIF2α phosphorylation (*Novoa et al., 2001*). In control cells (expressing the bland eGFP-PPP1R15B [1–146] targeting fragment), tunicamycin induced a transient and minor decrease in translation with a nadir at 2 hr (*Figure 9C*, lane 5). By contrast, in cells expressing ER-targeted mDia2, tunicamycin led to a sustained drop in protein synthesis associated with a sustained increase in eIF2α phosphorylation (*Figure 9C*, compare lanes 8 and 9). These experiments are consistent with a pool of G-actin localised in the vicinity of PPP1R15 in promoting eIF2α dephosphorylation.

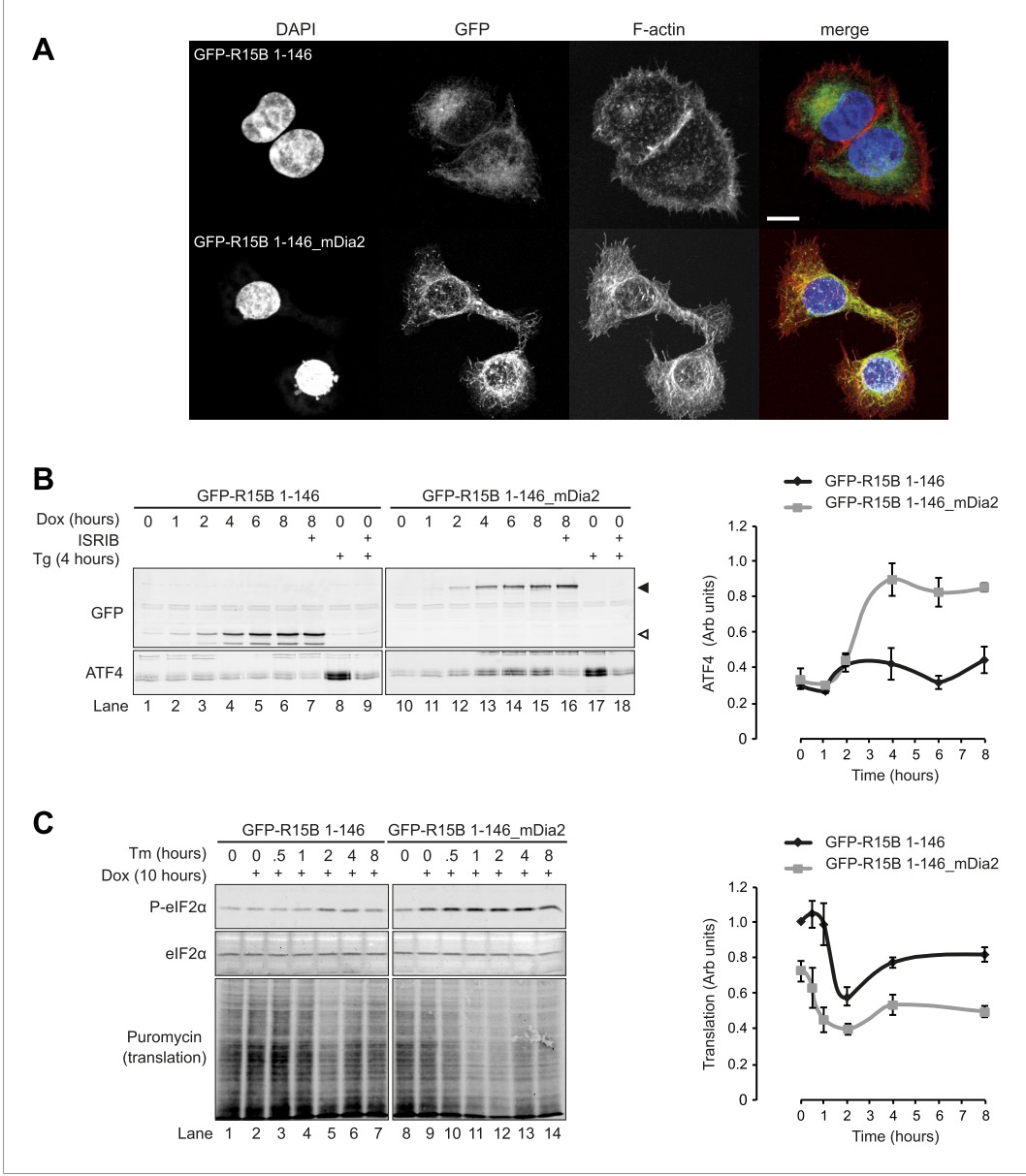

**Figure 9**. Localised changes in the polymeric status of actin modulate the sensitivity of the ISR. (**A**) Fluorescence microscopy image of Flp-In T-REx HEK293 cells treated with 1 µg/ml doxycycline for 12 hr to express either ER membrane-localised GFP (GFP-R15B 1–146) or ER membrane-localised GFP-mDia2 fusion (GFP-R15B 1–146_mDia2) then fixed and stained with Alexa-Fluor 568 phalloidin and imaged by confocal microscopy. Bar = 5 µm. (**B**) Immunoblot for GFP and ATF4 in lysates of GFP-R15B 1–146 or GFP-R15B 1–146_mDia2 Flp-In T-REx HEK293 cells following treatment with doxycycline (Dox) 0.1 µg/ml for indicated times or with ISRIB 100 nM and or thapsigargin 300 nM for 4 hr. Immunoreactivity to ATF4 was quantified using ImageJ software (ATF4 Intensity). Proteins of the expected sizes are marked with a solid triangle GFP-R15B 1–146_mDia2 or an open triangle GFP-R15B 1–146. (**C**) Immunoblot for P-eIF2α, total eIF2α, and puromycin in lysates of GFP-R15B 1–146 or GFP-R15B 1–146_mDia2 Flp-In T-REx HEK293 cells following pre-treatment—if indicated with doxycycline (Dox) 0.1 µg/ml for 10 hr followed by treatment with tunicamycin 2.5 µg/ml for indicated times. 10 min prior to harvesting, puromycin was added to the culture medium at a final concentration of 10 µg/ml. Immunoreactivity to puromycin within lysates served as a marker of protein translation and was quantified using ImageJ software (Puromycin intensity). Accompanying graphs of mean ± SEM of n = 3 independent repeats.

The following figure supplement is available for figure 9:

**Figure supplement 1**. Colocalisation of filamentous actin with ER membrane-localised GFP (GFP-R15B 1–146) or ER membrane-localised GFP-mDia2 fusion (GFP-R15B 1–146_mDia2).

## Discussion

Over the years multiple proteins have been noted to interact with the PPP1R15-PP1 core holoenzyme, but none has proved generalizable across experimental systems or successfully implicated in the genetically well-characterised role of the complex to promote eIF2α dephosphorylation (*Hasegawa et al., 2000a*, *2000b*; *Wu et al., 2002*; *Hung et al., 2003*; *Shi et al., 2004*). In this study, an unbiased approach identified actin as a conserved binding partner of PPP1R15. The affinities of actin for PPP1R15 lay within a physiologically relevant range such that fluctuations of the G:F actin ratio affected the amount of actin recovered in the complex. Alterations to the ratio of G:F actin at the site of PPP1R15 action were seen to modulate cellular sensitivity to ISR stimuli via changes in eIF2α phosphatase activity. Collectively, these findings establish G-actin as an important regulator of PPP1R15-mediated eIF2α dephosphorylation in vivo.

Our proteomics analysis also identified other potential binding partners of PPP1R15. In mammalian cells, tubulin and HSP70 were consistently recovered in complex with overexpressed PPP1R15 and PPP1R15-containing fusion proteins. These interactions are less conserved across phyla than the PPP1R15-actin interaction. Furthermore, in vitro experiments in the accompanying manuscript demonstrate that addition of actin is sufficient to endow the PPP1R15-PP1 complex with selectivity towards eIF2α (*Chen et al., 2015*). Thus, while there is nothing in our observations to argue against tubulin or HSP70 joining the complex and modulating PPP1R15-directed phosphatase activity, the evidence at hand suggesting actin's relevance to the core activity of the eIF2α-directed phosphatase justifies the focus on actin.

With polymerisation and depolymerisation, the actin cytoskeleton is highly dynamic and levels of G-actin are subject to large fluctuations. Following polymerisation of actin to the barbed end of a filament, bound ATP is hydrolysed and eventually ADP-actin dissociates from the pointed end (*Dominguez and Holmes, 2011*). This dynamic is regulated by proteins that enhance depolymerisation, for example, ADF, or promote the recharging with ATP, which enhances the recycling of monomers, for example, profilin (*Paavilainen et al., 2004*). Capping proteins prevent the consumption of monomers and so increase free G-actin concentrations, while severing proteins can lead to filament disassembly or nucleate more filament formation depending upon the context (*Wear and Cooper, 2004*). In contrast, formins like mDia2 remain associated with the barbed end yet promote addition of actin monomers.

Other actin-binding proteins have functions unrelated to the cytoskeleton and it is now well recognised that free G-actin can function as a second messenger. For example, MAL, a cofactor of the transcription factor SRF, cycles dynamically between the nucleus and cytoplasm in a manner regulated by its binding to G-actin in quiescent cells (*Miralles et al., 2003*; *Vartiainen et al., 2007*). By depleting G-actin, growth signal-driven actin polymerisation releases MAL to enter the nucleus, bind SRF and activate target genes. Other examples include Phactr, a PP1 regulatory subunit whose cytoplasmic localisation depends upon G-actin binding (*Wiezlak et al., 2012*) and RNA polymerases II and III for whom actin forms a scaffold for the assembly of enzyme complexes (*Hu et al., 2004*; *Kukalev et al., 2005*).

Many actin-binding proteins including MAL interact with a hydrophobic target-binding cleft between subdomains I and III of the actin monomer (*Mouilleron et al., 2008*; *Dominguez and Holmes, 2011*; *Shoji et al., 2012*). This site is blocked by cytochalasin D, which inhibits such interactions. Latrunculin B increases the level of actin monomers by binding to a different site on G-actin, the nucleotide-binding cleft, and does not interfere with binding at the hydrophobic cleft. Our observation that cytochalasin D diminishes the recovery of actin in complex with PPP1R15, is consistent with interaction via the hydrophobic target-binding cleft. While the precise details remain to be worked out, structural and biochemical studies presented in the accompanying manuscript support this idea and further suggest the C-terminal most residues of the functional core of the PPP1R15 family members play an important role in actin engagement (*Chen et al., 2015*). A crystal structure obtained for the binary complex of PPP1R15B and PP1 demonstrated that the N-terminal half of PPP1R15's functional core extensively engages the surface of PP1 following an itinerary previously observed for the regulatory subunit PPP1R9/spinophilin (*Ragusa et al., 2010*; *Chen et al., 2015*). Interestingly, the C-terminal portion of PPP1R15's functional core, implicated here in actin binding, was not observed in a high-resolution crystal structure of the PPP1R15B-PP1 binary complex, suggesting that this portion of PPP1R15B remained unstructured in the absence of actin. The crystal structure obtained for the 1:1:1 ternary complex of PPP1R15B-PP1-actin was of too low a resolution to identify these C-terminal residues of

PPP1R15's functional core, but unaccounted for density observed in the cleft between lobes I and III of actin suggests a mode of engagement of actin by this portion of PPP1R15B that would be sensitive to disruption by cytochalasin, which binds to the same region of G-actin.

Our in vivo findings reported here emphasize the importance of actin binding to the stability of the PPP1R15-PP1 complex and suggest that association of PP1 and actin with PPP1R15 may be cooperative. The accompanying manuscript provides further evidence for the direct binding of PPP1R15 and actin and reveals a role for actin in augmenting the specificity of the holophosphatase for eIF2α (*Chen et al., 2015*). These two mechanisms are likely to work in concert and suggest a crucial role for G-actin in establishing a biologically relevant route to eIF2α dephosphorylation.

It would appear that under normal circumstances G-actin is not limiting to eIF2α dephosphorylation in cultured MEFs, as latrunculin B, which enhances the pool of PPP1R15 binding-competent G-actin in some cell types, has no measurable effect on phosphorylated eIF2α (*Figure 5—figure supplement 1*). However, regulation of eIF2α phosphatases via the binding of G-actin might plausibly play a role in biological processes that are accompanied by changes in the ratio of G:F actin in other differentiated cell types, for example, in circumstances of cell migration, axonal guidance, or synaptic plasticity. The latter are particularly attractive given the evidence for crosstalk between the ISR and memory formation (*Costa-Mattioli et al., 2007*; *Ma et al., 2013*; *Sidrauski et al., 2013*).

The role of eIF2α phosphorylation in regulating rates of protein synthesis and the coupling of this phosphorylation event to the activation of a gene expression programme are conserved in eukaryotes. However, the mechanism for dephosphorylating eIF2α has diverged considerably. Yeasts rely on direct recruitment of the catalytic phosphatase subunit (Glc7p) to the eIF2 substrate, with no PPP1R15 intermediate (*Rojas et al., 2014*), while PPP1R15 family proteins are apparent only in complex animals: insects and vertebrates (*Novoa et al., 2001*; *Jousse et al., 2003*; *Malzer et al., 2013*). It is tempting to speculate that this more complex mode of regulating eIF2α dephosphorylation co-evolved with mechanisms for regulating the actin cytoskeleton and G-actin availability.

Current models suggest that PPP1R15B, which is expressed constitutively, provides a constant background of eIF2α phosphatase activity that is augmented by transcriptional induction of PPP1R15A during later stages of the ISR (*Jousse et al., 2003*). This study reveals that both PPP1R15 isoforms are poised to undergo post-translational regulation through changes in the polymeric status of actin. The focus here has been on the conserved functional core of PPP1R15, but there remains room for further modulation of both isoforms by their large, poorly characterised N-terminal regions. Our protein discovery effort has identified other interactors that may be unique to each isoform. Thus future studies to explore the possibility of differential regulation of eIF2α phosphatase activity by the different paralogues and their unique interactors seem warranted.

# Materials and methods

## Materials, plasmids, and primers

Jasplakinolide, thapsigargin, and tunicamycin were from Calbiochem (Millipore, Hertfordshire, UK), cytochalasin D was from Tocris (Bristol, UK), latrunculin B was from Enzo Life Sciences (Exeter, UK), Alexa Fluor 568 Phalloidin was from Life Technologies (Paisley, UK). PPP1R15ApEGFP-C3 and PPP1R15ApEGFP-N1 were kind gifts from S Shenolikar (Duke-NUS Graduate Medical School Singapore, Singapore) (*Zhou et al., 2011*). PerkKD-pGEX4T-1, dPPP1R15pEGFP, 2aOPTx3M(1–185)pET-30a(+), PPP1R15ApcDNA and dPPP1R15pEGFP have been described previously (*Harding et al., 1999*; *Novoa et al., 2003*; *Ito et al., 2004*; *Malzer et al., 2013*). PP1αpEBG was generated by ligating the human PP1α coding sequence into BamHI and NotI digested pEBG. For inducible HeLa cell lines, GFP-PPP1R15A was excised from PPP1R15ApEGFP-C3 with NheI and XhoI and ligated into NheI and SalI digested pTRE2Hyg (Clontech Laboratories, USA) to generate GFP-PPP1R15ApTRE2Hyg. PPP1R15A-GFP was excised from PPP1R15ApEGFP-N1 with BglII and NotI and ligated into BamHI and NotI digested pTRE2Hyg to generated PPP1R15A-GFPpTRE2Hyg. For PPP1R15B-GFP, PPP1R15BpEGFP-C1 was generated by ligating the human PPP1R15B coding sequence into BglII and SalI digested pEGFP-C1. For Flp-In T-REx HEK293 cell lines expressing GFP-R15B 1–146 and GFP-R15B 1–146_mDia2, the coding sequence for EGFP and residues 1–146 of human PPP1R15B was mobilized by digestion with NheI (partially repaired with Klenow-polymerase) and BamHI, before ligation into pcDNA5_TO_FRT (Life Technologies, USA) digested with HindIII (partially repaired with Klenow polymerase) and BamHI to generate EGFP_PPP1R15B_1–146pcDNA5_TO_FRT. PCR product encoding residues 532-1171 of

mDia2 was ligated into BamHI and XhoI digested EGFP_PPP1R15B_1–146 pcDNA5_TO_FRT to generate EGFP_PPP1R15B_1–146_mDia2_532-1171pcDNA5_TO_FRT. Primers used in this study are listing in *Table 1*.

## Site-directed mutagenesis

All truncations or point mutations in the PPP1R15A coding sequence were made as follows. Fifty nanograms of plasmid template DNA were mixed with 5 µl *Pfu turbo* DNA polymerase reaction buffer [10×], 1 µl *Pfu turbo* DNA polymerase (Agilent Technologies, Santa Clara, CA), 125 ng forward primer, 125 ng reverse primer, 1 µl of 25 mM dNTPs, made up to 50 µl with water. A PCR thermocycler was run using the following program parameters: 95°C for 30 s, 95°C for 30 s, 18 cycles (54°C for 1 min, 67°C for 20 min, 94°C for 1 min, 55°C for 1 min, 72°C for 10 min). Completed reactions were treated with 1 µl Dpn1 restriction enzyme, incubated at 37°C for 2 hr before using 5 µl of the reaction mix for a standard transformation into One Shot TOP10 chemically competent *E. coli* (Life Technologies, Paisley, UK).

## Cell culture

Mammalian cells, HEK293T, MEF (*Ppp1r15b*$^{tm1Dron/tm1Dron}$, *Ppp1r15a*$^{tm1Dron/tm1Dron}$, *Pkr–/–*, *Hri–/–*, *Perk–/–*, *Gcn2–/–*, *eIF2α*$^{AA}$), and NIH3T3, were maintained in DMEM supplemented with 10% vol/vol FBS and antibiotics (100U/ml Penicillin G and 100 µg/ml Streptomycin) and incubated at 37°C with 5% vol/vol $CO_2$ (*Yang et al., 1995*; *Harding et al., 2000*; *Han et al., 2001*; *Novoa et al., 2003*; *Scheuner et al., 2005*; *Harding et al., 2009*). HeLa Tet-On Advanced cells were purchased from Clontech Laboratories (Saint-Germain-en-Laye, France) and maintained in DMEM with 10% vol/vol tetracycline-free FBS and transfected with the expression vectors PPP1R15A-GFPpTRE2Hyg and GFP-PPP1R15ApTRE2Hyg. Stable clones were selected with 600 µM hygromycin. Transgene expression proved optimal when clones were treated with 1 µg/ml doxycycline.

## Immunoblots

Cell lysates were prepared in Harvest lysis buffer (HEPES pH 7.9, 10 mM; NaCl 50 mM; sucrose 0.5M; EDTA 0.1 mM; Triton X-100 0.5% vol/vol) supplemented with protease inhibitor cocktail (Roche, Welwyn Garden City, UK) and 1 mM PMSF. When analysing phospho-eIF2α, the lysis buffer was supplemented with phosphatase inhibitors (10 mM tetrasodium pyrophosphate, 15.5 mM β-glycerophosphate, 100 mM NaF). Cleared cell extracts were equalized by total cell protein using Bio-Rad protein assay (Bio-Rad, Hercules, CA, USA), boiled in SDS-loading buffer (25 mM Tris pH 6.8, 7.5% vol/vol glycerol, 1% wt/vol SDS, 25 mM DTT, 0.05% wt/vol bromophenol blue), subjected to reducing SDS-PAGE, and transferred to nitrocellulose membrane.

For GFP-Trap affinity purification, cells were lysed in the manufacturer's recommended buffers (Chromotek, Planegg-Martinsried, Germany) and incubated with GFP-Trap A beads according to manufacturer's instructions. Briefly, cells were lysed in GFP-Trap lysis buffer (150 mM NaCl, 10 mM Tris/Cl pH 7.5, 0.5 mM EDTA, 1 mM PMSF, and Protease Inhibitor Cocktail [Roche]) and post-nuclear supernatants were incubated with GFP-Trap beads at 4°C for 2 hr then washed four times in the same buffer. Proteins were eluted with SDS-PAGE loading buffer.

GST affinity purification was performed using Activated Thiol Sepharose 4B beads (GE Healthcare, Little Chalfont, UK). Briefly, cells were lysed with Harvest buffer, cleared by centrifugation and incubated with rotation with Activated Thiol Sepharose 4B beads for 2 hr at 4°C. Beads were then washed four times with lysis buffer and protein complexes then eluted by boiling with SDS-loading buffer or by addition of 20 mM glutathione.

For isolation of endogenous PPP1R15A by immunoprecipitation and PP1 by microcystin-affinity purification, MEF or HEK293T cells were lysed in lysis buffer (150 mM KCl, 20 mM HEPES pH 7.4, 2 mM $MgCl_2$, 1 mM PMSF, and Protease Inhibitor Cocktail) supplemented with either 0.1% (wt/vol) digitonin (Calbiochem, MERK Millipore, Darmstadt, Germany) or 0.5% (vol/vol) triton X-100, respectively. Immunoprecipitation of PPP1R15A was carried out for 16 hr at 4°C prior to three washes with detergent supplemented lysis buffer and elution in SDS-PAGE sample buffer. PP1 isolation by microcystin affinity purification was carried out for 1 hr at 4°C in the presence of 1 mM latrunculin B prior to three washes with detergent-supplemented lysis buffer and elution in SDS-PAGE sample buffer. Lysates were centrifuged at 200,000×*g* for 30 min to remove contaminating F-actin in order that G-actin levels were reflected in input samples.

**Table 1**. Primers used in this study

| hPPP1R15A truncation or point mutation | | | |
|---|---|---|---|
| 1–615 | For | | CCTGCTGCCCGGGCCAGAGCCTGAGCACGCCTCAGGAACC |
| | Rev | | GGTTCCTGAGGCGTGCTCAGGCTCTGGCCCGGGCAGCAGG |
| 1–620 | For | | CCTGGGCACGCCTCAGG**TAG**CCACCTTTAGCC |
| | Rev | | GGCTAAAGGTGGCTACCTGAGGCGTGCCCAGG |
| 501–654 | For | | TAA*GCTAGC*ACCATGGAAGCTGAGCCC |
| | Rev | | GTCGCGGCCGCTTTACTTGTACAGCTC |
| V[556]E | For | | CTAAAGGCCAGAAAGGAGCGCTTCTCCGAGAAGGTCACTG |
| | Rev | | CAGTGACCTTCTCGGAGAAGCGCTCCTTTCTGGCCTTTAG |
| W[616]A | For | | CCGGGCCAGAGCC**GC**GGCACGCCTCAGGAA |
| | Rev | | TTCCTGAGGCGTGCCGCGGCTCTGGCCCGG |
| R[618]A | For | | GCCAGAGCCTGGGCA**GC**CCTCAGGAACCCA |
| | Rev | | TGGGTTCCTGAGGGCTGCCCAGGCTCTGGC |
| L[619]A | For | | AGCCTGGGCACGC**GC**CAGGAACCCACCTTT |
| | Rev | | AAAGGTGGGTTCCTGGCGCGTGCCCAGGCT |
| R[620]A | For | | CCTGGGCACGCCTC**GC**GAACCCACCTTTAG |
| | Rev | | CTAAAGGTGGGTTCGCGAGGCGTGCCCAGG |
| W[616]A/L[619]A | For | | CGGGCCAGAGCC**GC**GGCACGC**GC**CAGGAACCC |
| | Rev | | GGGTTCCTGGCGCGTGCCGCGGCTCTGGCCCG |
| mDia2 fusion | For | | AATCCCGGATCCGTGCCTTGCCACCTGGTACA |
| | Rev | | AGCTCGCTCGAGTTATAAAGCTCGTAATCTTGCCAG |
| PPP1R15B fusion | | | |
| GFP fusion | For | | AGATT*AGATCT*GCCACCATGGAGCCGGGGACAGG |
| | Rev | | GATC*GTCGAC*ACATTGCTTGAGAACATTAAGTCC |
| dPPP1R15 truncation | | | |
| 1–307 | For | | CCAGTTCACCGAGATCGTTAGTACCAAGCTCGATTCTTGCACG |
| | Rev | | CGTGCAAGAATCGAGCTTGGTACTAACGATCTCGGTGAACTGG |
| 1–312 | For | | CGTGTCTACCAAGCTCGATAGTTGCACGAGGACGAGC |
| | Rev | | GCTCGTCCTCGTGCAACTATCGAGCTTGGTAGACACG |

Primary antibodies used were: rabbit anti-PPP1R15A (10,449-1-AP, 1:1000; Proteintech, Manchester, UK) mouse anti-GFP antibody (ab1218, 1:1000; Abcam, Cambridge, UK), rabbit anti-PP1α antibody (no. 2582, 1:1000; Cell Signaling, Danvers, MA, USA), rabbit p-eIF2α (3597, Cell Signaling; 1:1000), anti-actin (ab3280, 1:1000; Abcam), rabbit anti-ATF4 (C-20, 1:500; SantaCruz, Santa Cruz, CA, USA), mouse anti-puromycin antiserum (PMY-2A4, Developmental Studies Hybridoma Bank, University of Iowa, USA), anti-total eIF2α mouse monoclonal (AHO0802,1:1000; Invitrogen, Thermo Fisher Scientific, Waltham, MA, USA).

## Coomassie and silver stain

Gels were stained with InstantBlue Coomassie stain (Expedeon, San Diego, CA, USA) as directed by the manufacturer's instructions. For silver staining: gels were fixed (10% vol/vol methanol, 7.5% vol/vol acetic acid) for 20 min with agitation followed by two quick rinses with water. Gels were then incubated with 3.25 µM DTT in water for 20 min with agitation then 0.1% wt/vol $AgNO_3$ in water for 30 min with agitation. Following a 1-min wash with water, gels were developed using 3% wt/vol $Na_2CO_3$, 0.02% wt/vol formaldehyde in water until bands became visible and the reaction was stopped with fixative.

## F-actin sedimentation assay

HEK293T cells were transfected with PP1αpEBG and untagged PPP1R15ApcDNA. After 24 hr, cells were lysed in harvest buffer and subjected to GST affinity purification. Protein complexes were eluted with 20 mM reduced glutathione in 50 mM Tris pH 7.5. The eluate was mixed with 10 µM purified F-actin in actin binding buffer (20 mM Tris pH 8, 100 mM NaCl, 2 mM $MgCl_2$, 1 mM ATP, 1 mM DTT, 0.1 mM $CaCl_2$) in a total volume of 200 µl. Samples were centrifuged at 279,000×$g$ for 15 min in a TLA120.1 rotor. The supernatant was removed and mixed with 50 µl of 4× SDS loading buffer, while the pellet was re-suspended in 250 µl of 1× SDS loading buffer. Samples were then boiled and analysed by SDS-PAGE.

## In vitro eIF2α de-phosphorylation assay

Phosphorylated recombinant eIF2α N-terminal domain (NTD) was generated as described previously (*Marciniak et al., 2006*). The expression plasmid PerkKD-pGEX4T-1 encoding GST-PERK kinase domain fusion protein of mouse PERK residues 537–1114 wild type has previously been described (*Harding et al., 1999*). eIF2α-NTD encoding residues 1–185 of human eIF2α with three solubilizing mutations was bacterially expressed from codon optimized vector 2aOPTx3M(1–185)pET-30a(+) (*Ito et al., 2004*). Bacterially expressed GST-PERK immobilised on activated thiol sepharose beads was incubated with 10 µl of 1 mM ATP and bacterially expressed eIF2α-NTD at 37°C with shaking in 20 µl kinase buffer (5×: 100 mM TRIS pH 7.4, 250 mM KCl, 10 mM Mg(OAc)$_2$, 10 mM $MnCl_2$, and 5 mM DTT) made up to 100 µl total reaction volume. GST-PERK beads were removed by centrifugation and remaining ATP was removed by dialysis against reaction buffer. The resulting phosphorylation eIF2α-NTD was incubated with affinity-purified phosphatase in 20 mM Tris HCL pH 7.4, 50 mM KCl, 2 mM $MgCl_2$, 0.1 mM EDTA, 0.8 mM ATP at 30°C for indicates times with shaking. Reactions were terminated by the addition of Laemmli buffer.

## Immunofluorescence microscopy

Cells plated onto glass coverslips were washed twice with PBS and fixed with 4% formaldehyde for 20 min. Following a further two PBS washes, cells were then permeabilised with 0.5% vol/vol triton X-100 in PBS for 3 min then blocked with 1% wt/vol BSA in PBS for 1 hr. Cells were then incubated in the dark with Alexa-Fluor 568 phalloidin 1:40 for 1 hr. After three 5-min washes in PBS, the glass coverslips were mounted onto slides using ProLong Gold antifade reagent (Life Technologies) ready for visualisation.

## Acknowledgements

This work was funded by the Medical Research Council (UK) (MRC Ref G1002610) and a Wellcome Trust Strategic Award for core facilities to the Cambridge Institute for Medical Research (CIMR, Wellcome 100140). SJM holds a Senior Clinical Research Fellowship from the Medical Research Council (MRC Ref G1002610). DR is a Wellcome Trust Principal Research Fellow (Wellcome 084812/Z/08/Z). The June Hancock Mesothelioma Research Fund funded LED (JH09-2); the British Lung Foundation funded HJC (APHD11-4); CD is a member of the CIMR PhD programme funded by the Wellcome Trust; and VP holds a Diabetes UK Arthur and Sadie Pethybridge PhD Studentship. We are grateful to William Meadows and Roger B Dodd (University of Cambridge, UK) for advice and technical assistance in mammalian cell culture.

# Additional information

## Competing interests

DR: Reviewing editor, *eLife*. The other authors declare that no competing interests exist.

## Funding

| Funder | Grant reference | Author |
|---|---|---|
| Medical Research Council (MRC) | G1002610 | Stefan J Marciniak |
| Wellcome Trust | 084812/Z/08/Z | David Ron |
| British Lung Foundation (BLF) | APHD11-4 | Hanna J Clarke |
| June Hancock Mesothelioma Research Fund | JH09-2 | Lucy E Dalton |
| Wellcome Trust | 100140 | David Ron |
| Diabetes UK | 12/0004595 | Vruti Patel |
| Wellcome Trust | CIMR PhD programme | Caia S Dominicus |

The funders had no role in study design, data collection and interpretation, or the decision to submit the work for publication.

## Author contributions

JEC, Performed experiments including generation and use of stables cells for tetracycline-inducible expression of mDia2 fusion proteins, in vivo assessment of eIF2α phosphatase activity by use of eIF2α kinase inhibition and contributed to the writing of the manuscript., Acquisition of data, Analysis and interpretation of data, Drafting or revising the article; LED, Performed experiments including GFP-affinity purification of PPP1R15A and mass spectroscopy, deletion mapping for in vivo actin interactions, manipulation of actin polymerisation status by drugs and serum, in vitro eIF2α phosphatase assay, generated stable cells for tetracycline-inducible expression of PPP1R15A, and contributed to the writing of the manuscript., Acquisition of data, Analysis and interpretation of data, Drafting or revising the article; HJC, Performed experiments including GFP-affinity purification of PPP1R15A for mass spectroscopy, examined effect of jasplakinolide in eIF2α kinase knockout lines, manipulation of actin polymerisation status by drugs and serum, and contributed to the writing of the manuscript., Acquisition of data, Analysis and interpretation of data, Drafting or revising the article; EM, Perform affinity purification and deletion mapping experiments involving dPPP1R15, and contributed to the writing of the manuscript., Acquisition of data, Analysis and interpretation of data, Drafting or revising the article; CSD, Performed experiments including GFP-affinity purification of PPP1R15B for mass spectroscopy, and contributed to the writing of the manuscript., Acquisition of data, Analysis and interpretation of data, Drafting or revising the article; VP, Examined the effect of manipulating actin polymerisation status by drugs and contributed to the writing of the manuscript., Acquisition of data, Analysis and interpretation of data, Drafting or revising the article; GM, Greg Moorhead synthesised and provided microcysin agarose. The labour-intensive production of this reagent (which is not available commercially) enabled a key experiment. He also contributed his expertise in phosphatase biology to review and approve the final version of the manuscript., Drafting or revising the article, Contributed unpublished essential data or reagents; DR, Contributed to experimental strategies, designed mDia2 constructs and contributed to the writing of the manuscript., Acquisition of data, Analysis and interpretation of data, Drafting or revising the article; SJM, Conceived and oversaw the study as a whole, discovered the interaction between PPP1R15A and actin, and wrote the manuscript., Conception and design, Analysis and interpretation of data, Drafting or revising the article

## Author ORCIDs

David Ron, http://orcid.org/0000-0002-3014-5636
Stefan J Marciniak, http://orcid.org/0000-0001-8472-7183

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
