## [Decision Letter]

Thank you for sending your work entitled ”Actin dynamics tune the integrated stress response by regulating eIF2α dephosphorylation“ for consideration at *eLife.* Your article has been evaluated by Vivek Malhotra (Senior editor), a Reviewing editor, and three reviewers, one of whom, Richard Treisman, has agreed to reveal his identity.

All three reviewers, with different degrees of enthusiasm, thought that the work in this manuscript was interesting and potentially important. The Reviewing editor and the other reviewers discussed their comments before we reached this decision, and the Reviewing editor has assembled the following comments to help you prepare a revised submission.

The work shows that G-actin interacts with various R15 isoforms, and the data in this paper together with the data in the accompanying paper (Chen et al., 2015, *eLife*) support a possible model where the inclusion of G-actin in a trimeric complex is important for substrate specificity. The work is based on a fairly straightforward workflow involving protein pull-down studies, protein identification by mass spectrometry, structure-function analysis to define regions and residues that interact, then a more functional set of studies where G to F-actin dynamics are manipulated in cells. Overall the study is technically solid and the results are of potential importance. 

Substantive concerns:

1) The pull-down experiments identify the most abundant proteins in cells, and while a number of controls are included, there is a concern that the work relies on over-expression of proteins in model cell preparations. As the authors acknowledge in the Discussion, the artificial manipulations used to perturb G-actin, do not really inform upon any biological process where G- to F-actin dynamics are altered.

2) There is little information related to replication of experiments. Representative studies are presented but little cumulative data is analyzed and quantified.

3) A commercial GFP pull-down kit is used. It is important that the buffer conditions are included in the Methods section. In this regard, was G-actin binding sensitive to lysis or other buffer components including salt and detergent?

4) The mass spectrometry data is not presented.

5) Relative stability of trimeric complex: the authors’ assertion that the ternary complex is more stable than binary complexes (at the end of the subsection “Actin associates with the conserved C-terminal functional core of PPP1R15”) seems to be based on the analysis of the V^556^E mutant, but no evidence in this experiment that addresses this issue directly.

6) In Figure 1, tubulin also seems to be a conserved PPP15A/B interactor, and in Figure 7 the amount of tubulin associated w/ the PPP15A-GFP complex is also decreased when purified from jasplakinolide treated cells. It would be helpful to know (and potentially a great control) if eIF2α phosphorylation is altered in cells treated with drugs that alter levels of MT polymerization. In addition, some comment on the tubulin recovery in the experiments would be appropriate; what do the authors think is happening here?

7) Since jasplakinomide perturbs actin polymerization through a specific mechanism, it is important to use another method (for example phalloidin) to ensure that the effects are not specific to jasplakinomide.

8) Figure 4: please indicate the time of treatment.

9) Figure 4: while a control experiment is shown in Figure 4, it is hard to make much of a conclusion since the result in Figure 4 appears to be fully explained by the level of induced expression of R15A. In addition, in Figure 4 an important control is not shown: overall levels of eIF2α. Its absence is conspicuous as this control is shown for other analyses, including panel 4C.

10) Figure 6: the data for the R15A mutant cells seems non-informative since little R15A is expressed at baseline.

11) Figure 7: this experiment is not explained clearly in the text (subsection headed “Association of G-actin with PPP1R15 regulates eIF2α phosphatase activity in vivo”).

12) Figure 8: do the constructs with the membrane-targeting domain of R15b displace the ER location of R15B or R15A? In addition, the cells in appear significantly different in morphology in this figure. Please show more examples, and provide some quantitation of the extent of colocalization of F-actin and GFP.

13) Referencing: the use of the Sotiropoulos reference for MAL and actin is wrong. 1) Initial finding should be Miralles et al., Cell 2003 vol. 113, 329; 2) Rather than being held in the cytoplasm by actin and released for nuclear entry, MAL is actually dynamic, with the protein continually circulating through the nucleus ([47] Science vol. 316, 1749. 3. Structural analysis of binding to actin was initially done by [32] EMBO 27, 3198).

[Editors' note: further revisions were requested prior to acceptance, as described below.]

Thank you for resubmitting your work entitled ”Actin dynamics tune the integrated stress response by regulating eukaryotic initiation factor 2α dephosphorylation“ for further consideration at *eLife*. Your revised article has been favorably evaluated by Vivek Malhotra (Senior editor) and a member of the Board of Reviewing Editors.

Two of the original three reviewers examined the revised manuscript and one of them, self-identified as Richard Treisman, found the revisions appropriate and had only one small criticism that you need to address.

The other reviewer had more substantive criticisms, many of which he/she felt were raised in the first round of reviews, that we have listed below. Specifically, he/she remains concerned about the stoichiometry of the PP1/R15 complex and actin. He/she notes that this interaction is at the core of the work and conclusions, and is not convinced from the data that the interaction is physiologically significant. This issue, therefore, remains a significant problem that needs addressing directly. The reviewer suggests, as did on the original reviews, that purified proteins would be a better basis for this analysis.

Reviewer #1 (Richard Treisman):

Point 6 in the rebuttal: “OK, but puzzled as to the effect on Tubulin recovery. The authors should add a comment as to what they think is going on here.”

Reviewer #2 (verbatim):

The authors have made a reasonable attempt to address the reviewers' points in the revised version of this manuscript. There were concerns that the approach used overexpression of proteins in model cell systems, that G-actin and the other proteins (tubulin, HSP70) were highly abundant and that the perturbations to G-actin/F-actin were non-physiological. While additional studies have been performed to examine endogenous PPP1R15 and PP1, the results raise some additional questions, and many of the original concerns remain.

Specific comments:

1) It is a concern that actin cannot be detected in the pulldown studies added in Figure 4. There is a vague comment about the antibody not being effective enough, but this does not make a lot of sense, given that there is a supposedly ”stoichiometric“ amount of actin coming down with the PP1/R15 complex (Figure 1 and Results). If this is the case, then it should be feasible through the use of internal standards to work out how much endogenous PP1 and R15 are isolated relative to the amounts of PP1 and R15 isolated following over-expression, then to set up an assay to measure comparable amounts of actin. Is there still tubulin and HSP70 pulled down with endogenous PP1 and R15? If so, and no actin can be detected, then how does one interpret the physiological significance of the proposed G-actin interaction.

2) Related to this point of stoichiometry, if there is a stoichiometric amount of G-actin bound to PP1/R15 then what does this say about how tubulin or HSP70 might also be binding to the complex?

3) The questions related to the binding between the different proteins really needs evidence using purified components, which is the topic of the accompanying paper ([4], *eLife*). In isolation it is not easy to judge the potential physiological significance of the data presented in this paper, and the Discussion really needs to make it more clear how the structural data in the other paper help interpret the data in this paper.

4) Abstract: the data presented here are more supportive of G-actin being able to stabilize the PP1/R15 complex rather than being an ”activator”.

---

## [Author Response]

*1) The pull-down experiments identify the most abundant proteins in cells, and while a number of controls are included, there is a concern that the work relies on over-expression of proteins in model cell preparations. As the authors acknowledge in the Discussion, the artificial manipulations used to perturb G-actin, do not really inform upon any biological process where G- to F-actin dynamics are altered*.

We have now immunoprecipitated endogenous PPP1R15A from cell lysates and demonstrated the striking dependence of PP1 binding to PPP1R15A on G-actin abundance within the cell (new Figure 4). Due to poor sensitivity of the available antisera to actin we were unable to resolve associated actin from background, and so we depleted input lysate of F-actin by centrifugation in order to report G-actin in the input samples. It is clearly seen, that in samples depleted of soluble G-actin, less PP1 is recovered in complex with endogenous PPP1R15A (compare lanes 2 and 3, Figure 4). As a complementary approach, we used microcystin-agarose matrix to purify endogenous PP1-containing complexes from cell lysates and observed that depletion of G-actin reduced the quantity of endogenous PPP1R15A associated with the catalytic subunit (new Figure 4).

The following has been added to the main text of the manuscript:

“To examine the relevance of G-actin to the endogenous PPP1R15 complex, wild type Ppp1r15a^+/+^ and mutant Ppp1r15a^mut/mut^ mouse embryonic fibroblasts (MEFs) […] These findings are consistent with a role for G-actin in promoting association between endogenous PPP1R15A and PP1.”

And to the Abstract:

“Like PP1, G-actin associated with the functional core of PPP1R15 family members and G-actin depletion, by the marine toxin jasplakinolide destabilised the endogenous PPP1R15A-PP1 complex.”

*2) There is little information related to replication of experiments. Representative studies are presented but little cumulative data is analyzed and quantified*.

We now include the quantification of three independent replicates of Figure 5, Figure 6, Figure 7, Figure 8, Figure 9.

*3) A commercial GFP pull-down kit is used*. *It is important that the buffer conditions are included in the Methods section. In this regard, was G-actin binding sensitive to lysis or other buffer components including salt and detergent?*

The Methods have be amended as follows:

“For GFP-Trap® affinity purification, cells were lysed in the manufacturer’s recommended buffers (Chromotek, Germany) and incubated with GFP-Trap® A beads according to manufacturer’s instructions. Briefly, cells were lysed in GFP-Trap lysis buffer [150mM NaCl, 10 mM Tris/Cl pH 7.5, 0.5 mM EDTA, 1 mM PMSF and Protease Inhibitor Cocktail (Roche)] and post-nuclear supernatants were incubated with GFP-Trap beads at 4°C for two hours then washed four times in the same buffer. Proteins were eluted with SDS-PAGE loading buffer.”

The lysis buffers routinely contained physiological levels of salt, but as suggested we have now analysed the interactions between PPP1R15A and either actin or PP1 (new Figure 3—figure supplement 1). Lysis buffers routinely contained no more than 1% v/v Triton X-100 and beads wash buffers were detergent-free. In new Figure 3—figure supplement 2, we demonstrate that the interaction between PPP1R15A and both actin and PP1 were sensitive to detergent. When lysates were subjected to anti-GFP immunoprecipitation and then washed either without detergent, with Triton X-100, RIPA (1% NP40, 0.5% sodium deoxycholate, 0.1% SDS) or with digitonin-containing buffers, all except RIPA preserved the R15A-actin and R15A-PP1 associations. This suggests that while the complex is stable in a number of non-denaturing detergents, denaturing detergents may not be compatible with it. The following has been added to the Results section:

“The quantities of actin and PP1 recovered in complex with PPP1R15A were sensitive to the salt concentration of the buffers used (Figure 3—figure supplement 1). […] RIPA, which contains the denaturing detergents sodium deoxycholate (0.5% v/v) and SDS (0.1% v/v), completely abolished interaction between PPP1R15A and both PP1 and actin (Figure 3—figure supplement 2).”

*4) The mass spectrometry data is not presented*.

We have included total spectra count, total unique peptides and percentage coverage for GFP, GFP-PPP1R15A, GFP-PPP1R15b, V5 and V5-dPPP1R15 in new Figure 1—figure supplement 1 and Figure 1—figure supplement 2.

*5) Relative stability of trimeric complex: the authors’ assertion that the ternary complex is more stable than binary complexes (at the end of the subsection “Actin associates with the conserved C-terminal functional core of PPP1R15”) seems to be based on the analysis of the V*^*556*^*E mutant, but no evidence in this experiment that addresses this issue directly*.

We agree that while deletion and point mutants support a measure of independent binding of actin and PP1 to PPPR15 the new data from purification of endogenous PPP1R15A complexes indicate that manipulation of G-actin availability directly affects the association of endogenous PP1 with endogenous PPP1R15A (new Figure 4). It can also be seen that PPP1R15A isolated from jasplakinolide-treated cells binds far less PP1 and so is significantly less effective at dephosphorylating eIF2α in vitro (Figure 8). Furthermore the new Figure 3—figure supplement 1 confirms that the V^556^E mutant of PPP1R15A, which is unable to engage PP1 stably, also interacts poorly with actin across a broad range of salt concentrations, in marked contrast to wild type PPP1R15A. We have therefore amended the text as follows:

“Mutational analysis thus indicates that, while independent association of PP1 or actin with PPP1R15 may exist, the ternary complex of wild type PPP1R15, actin and PP1 is more readily recoverable.”

*6) In*
Figure 1*, tubulin also seems to be a conserved PPP15A/B interactor, and in*
Figure 7
*the amount of tubulin associated w/ the PPP15A-GFP complex is also decreased when purified from jasplakinolide treated cells. It would be helpful to know (and potentially a great control) if eIF2α phosphorylation is altered in cells treated with drugs that alter levels of MT polymerization. In addition, some comment on the tubulin recovery in the experiments would be appropriate; what do the authors think is happening here?*

We agree that the potential for involvement of tubulin in mammalian PPP1R15-containing phosphatases is potentially interesting. However, in the current manuscript, we decided to focus on the phylogenetically conserved interaction of the R15s with actin, rather than on tubulin, which has a far less strong interaction with dPPP1R15.

In reference to the reviewers’ logical suggestions for further experimentation, we note that manipulation of microtubules to increase their polymerization (e.g. with taxanes) leads to increased eIF2α phosphorylation through effects on the kinase PERK (Swanton et al., 2007 Cancer Cell 11: 498-512). This imposes further challenges on isolation of effects of manipulating the microtubule network on the eIF2⍺ phosphatases and is an issue that may need to be addresses by future studies.

*7) Since jasplakinomide perturbs actin polymerization through a specific mechanism, it is important to use another method (for example phalloidin) to ensure that the effects are not specific to jasplakinomide*.

To avoid confounding off-pathways effects of small molecules, we generated stable cells conditionally expressing a construct comprising constitutively active mDia2 fused to the membrane-associating portion of PPP1R15B (residues 1-146). mDia2 is a formin that both nucleates and processively elongates actin filaments. In Figure 9 and Figure 9—figure supplement 1 it can be seen that expression of this construct led to a marked increase in F-actin within the cell in a central distribution rather than the cortical arrangement observed in control cells. This increase in F-actin, which showed significant co-localisation with GFP-R15B 1-146_mDia2 (Figure 9—figure supplement 1), was associated with induction of the integrated stress response effector ATF4. The complete abrogation of this by ISRIB confirmed that induction of ATF4 by expression of the mDia2 construct was mediated by phosphorylation of eIF2α (compare lanes 15 and 16 of Figure 9). Moreover, induction of the mDia2 construct was sufficient to cause phosphorylation of eIF2α (compare lanes 8 and 9 of Figure 9).

The reviewers’ suggestion that we exploit phalloidin to alter actin polymerization status is sensible. However, while this toxin indeed stabilizes filaments of F-actin, it is poorly membrane permeant and difficult to use in the bulk-based endpoint assays available to monitor the ISR, eIF2⍺ phosphorylation and PPP1R15 complex composition. Membrane disruption by scraping (“scrape loading”) has been described as a means to allow it to gain access to the cell (Serpinskaya et al., 1990, FEBBS Lett 17: 11-14). In our hands, when applied to the culture media, phalloidin failed to induce any morphological features of actin polymerization, while scrape loading alone induced significant confounding cellular stress. Phalloidin has also been introduced into cells by microinjection (Wehland and Weber, 1981**,** Eur J Cell Biol 24, 176–183), but this does not permit the analysis of sufficient numbers of cells for immunoblot. The cell permeant semi-synthetic derivative, phalloidin-oleate, is no longer commercially available.

*8)*
Figure 4*: please indicate the time of treatment*.

The incubation was for 1 hour and the legend has been amended as follows, and this figure is now Figure 5:

“Figure 5. Association of G-actin with PPP1R15 regulates eIF2α phosphatase activity… (B) WT MEFs were treated with the indicated concentrations of jasplakinolide for 1 hour. Lysates were analysed as in (A)…”

*9)*
Figure 4*: while a control experiment is shown in*
Figure 4*, it is hard to make much of a conclusion since the result in*
Figure 4
*appears to be fully explained by the level of induced expression of R15A. In addition, in*
Figure 4
*an important control is not shown: overall levels of eIF2α. Its absence is conspicuous as this control is shown for other analyses, including panel 4C*.

As requested, we have now included the total eIF2α control (amended Figure 5; note number change owing to new Figure 4).

We remain of the opinion that overexpression of PPP1R15A is a valid means to isolate the effect of jasplakinolide on eIF2α phosphorylation from its effects on PPP1R15A expression. Differences in R15A induction are likely to reflect differences in cap-dependent translation between the two conditions and so in Figure 5 (previously 4D) we over-expressed PPP1R15A from the outset of the experiment thus equalizing its levels between control and jasplakinolide-treated cells. In vehicle-treated cells, a substantial reduction of eIF2α phosphorylation was observed owing to increased eIF2α phosphatase activity, while in jasplakinolide-treated cells eIF2α phosphorylation was dramatically increased consistent with reduction of eIF2α phosphatase activity. We address and exclude the possibility of these effects resulting from altered eIF2α kinase activity elsewhere in the manuscript (Figure 6). The main text has been amended as follows:

“In the presence of jasplakinolide, the elevated levels of phosphorylated eIF2α induced by thapsigargin persisted (Figure 5 lanes 7-12) […] However, jasplakinolide reversed the inhibitory effect of PPP1R15A over-expression restoring elevated levels of phosphorylated eIF2α to thapsigargin-treated cells (Figure 5).”

*10)*
Figure 6*: the data for the R15A mutant cells seems non-informative since little R15A is expressed at baseline*.

Whilst at baseline there are low (but nonetheless detectable) levels of PPP1R15A (new Figure 4), to isolate the effect of PPP1R15B from PPP1R15A we conducted these experiments in a PPP1R15A functional null cell (Figure 7, previously 6C and D). These experiments were important to establish the sensitivity of PPP1R15B to actin polymerization. We have amended the text as follows to clarify this point:

“Enhanced levels of phosphorylated eIF2α in jasplakinolide-treated cells and the synergistic effects of depleting G actin on the response to thapsigargin were observed in wildtype cells and in cells lacking either PPP1R15A or PPP1R15B-directed eIF2α dephosphorylation (Figure 6). These observations indicate that G-actin plays a functional role in holophosphatases constituted with either regulatory subunit. Importantly, this is the first evidence that the activity of constitutively expressed PPP1R15B can be modulated.”

*11)*
Figure 7*: this experiment is not explained clearly in the text (subsection headed “Association of G-actin with PPP1R15 regulates eIF2α phosphatase activity in vivo”)*.

We have amended the text as follows (note that Figure 7 is now Figure 8 in the resubmitted manuscript):

“To explore in further detail the basis for the correlation between G-actin levels and eIF2α dephosphorylation […] indicating that stabilisation of a complex between PPP1R15 and PP1 dominates G-actin’s role in this experimental system.”

*12)*
Figure 8*: do the constructs with the membrane-targeting domain of R15B displace the ER location of R15B or R15A? In addition, the cells in appear significantly different in morphology in this figure. Please show more examples, and provide some quantitation of the extent of colocalization of F-actin and GFP*.

As requested, we now include additional images of cells expressing each construct and quantification of the co- localization of F-actin and GFP (Figure 9—figure supplement 1).

Since both the control and mDia2 constructs were directed to endomembrane by the R15B N-terminal domain, the possibility that these might displace endogenous R15 exists. This requires the R15 binding site to be saturable, but we have no evidence for the latter. Indeed, overexpression of GFP-tagged N-terminal domain fails to affect mCherry-R15 localization (not shown). However, the hypothesis we set out to test in the experiments of Figure 9 was that depletion of a local pool of actin adjacent to the ER would enfeeble R15 activity. If the control construct were to deplete R15 from its normal location and if this were to impair its enzymatic activity, the observed effect of mDia2 induction would be minimized since eIF2α phosphatase activity would have been impaired similarly by the GFP-R15B 1-146 control construct (expressed at similar levels to the mDia2 construct on western blots). In this regard, our observation that GFP-R15B 1-146_mDia2 leads to activation of ATF4 is strengthened. We now address this in the Discussion as follows:

“In an effort to examine the effect of localized polymerization of actin, while simultaneously overcoming our concern of off-pathway effects […] our observation that GFP-R15B 1-146_mDia2 leads to activation of ATF4 is strengthened.”

*13) Referencing: the use of the Sotiropoulos reference for MAL and actin is wrong. 1) Initial finding should be Miralles et al., Cell 2003 vol. 113, 329; 2) Rather than being held in the cytoplasm by actin and released for nuclear entry, MAL is actually dynamic, with the protein continually circulating through the nucleus (*[47]
*Science vol. 316, 1749. 3. Structural analysis of binding to actin was initially done by*
[32]
*EMBO 27, 3198)*.

We thank the reviewer for the suggestions and have amended the citations accordingly.

[Editors' note: further revisions were requested prior to acceptance, as described below.]

*The other reviewer had more substantive criticisms, many of which he/she felt were raised in the first round of reviews, that we have listed below. Specifically, he/she remains concerned about the stoichiometry of the PP1/R15 complex and actin. He/she notes that this interaction is at the core of the work and conclusions, and is not convinced from the data that the interaction is physiologically significant. This issue, therefore, remains a significant problem that needs addressing directly. The reviewer suggests, as did on the original reviews, that purified proteins would be a better basis for this analysis*.

The strongest functional evidence for the role of G-actin in regulating the phosphatase activity of PPP1R15 comes from the various experiments in which manipulation of the actin cytoskeleton in living cells is shown to modulate eIF2α dephosphorylation (Figures 5, 6, 7 and 9).These correlate with powerful negative effects of depletion of G-actin on the stability of the endogenous PPP1R15-PP1 complex recovered from cells (Figures 4 and 8). As the editor and reviewer 2 noted, the issue of stoichiometry is best addressed in vitro using purified components. This indeed has been done in the accompanying manuscript ([4], *eLife*), revealing a 1:1:1 complex of PPP1R15-PP1-actin that co-purifies and crystalizes. This point and its significance are now discussed in detail in the revised version of manuscript (in the Discussion).

Because of its importance to this paper we have further probed the actin content of the endogenous complex of PPP1R15-PP1. We faced three challenges: (1) poor recovery of endogenous PPP1R15 by immunoprecipitation, (2) poor reactivity of the available anti-actin sera, and (3) actin contamination of reference immunoprecipitations (the controls). Problem 3 proved especially prominent in mouse fibroblasts (MEFs), but could be circumvented partially by using HEK293T cells. Therefore, we first confirmed the crucial role of the actin cytoskeleton on the stability of the PPP1R15-PP1 complex in HEK293T cells by comparing the recovery on microcystin beads of endogenous PPP1R15A from untreated and jasplakinolide-treated cells (an assay for the stability of the PPP1R15-PP1 complex, which associates with the microcystin via its PP1 component) (new Figure 4). Then, following the reviewers’ suggestion, we scaled up the input material for the immunoprecipitation to recover endogenous PPP1R15A at a quantity comparable to that of the over-expressed PPP1R15A and assayed for the associated actin. As shown in new Figure 4, comparable levels of actin were found, supporting the conclusion that actin is a functionally relevant constituent of the endogenous PPP1R15-containing complex.

Reviewer #1 (Richard Treisman):

*Point 6 in the rebuttal: ”OK, but puzzled as to the effect on Tubulin recovery. The authors should add a comment as to what they think is going on here*.*“*

We have amended the Discussion as follows:

“Our proteomics analysis also identified other potential binding partners of PPP1R15. […] the evidence at hand suggesting actin’s relevance to the core activity of the eIF2α-directed phosphatase justifies the focus on actin.”

Reviewer #2 (verbatim):

*1) It is a concern that actin cannot be detected in the pulldown studies added in*
Figure 4*. There is a vague comment about the antibody not being effective enough, but this does not make a lot of sense, given that there is a supposedly ”stoichiometric“ amount of actin coming down with the PP1/R15 complex (*Figure 1
*and Results). If this is the case, then it should be feasible through the use of internal standards to work out how much endogenous PP1 and R15 are isolated relative to the amounts of PP1 and R15 isolated following over-expression, then to set up an assay to measure comparable amounts of actin*.

We thank the reviewer for this suggestion. As noted previously, new Figure 4 presents an analysis of the recovery of endogenous actin in complex with endogenous PPP1R15A-PP1 and a side-by-side comparison to the recovery of actin in complex with over-expressed PPP1R15A. For this, it proved necessary to use HEK293T cells (rather than MEFs which had prohibitive non-specific binding of actin in the control immunopurification samples).

The new experiment reveals comparable association of actin with endogenous and over-expressed PPP1R15A and thus supports the extensive functional experiments documenting the dependence of eIF2α dephosphorylation on the availability of G-actin.

Two further points are worth making in regards to this experiment: (1) The exposure of cells to jasplakinolide markedly increased the non-specific recovery of actin in control immunoprecipitation samples therefore it was not possible to measure the effect of jasplakinolide on the recovery of actin in complex with PPP1R15A. (2) Similar considerations apply to the recovery of actin on microcystin beads, the latter may be due to the known existence of other PP1-actin complexes in the cell (22; 36).

The following new text has been added to the Results section under the subsection headed “Association of G-actin with PPP1R15 regulates eIF2α phosphatase activity in vivo”:

“Whilst the presence of other known PP1-actin complexes precludes meaningful interpretation of actin purified by microcystin affinity […] This supports a role for the interaction in cell physiology.”

*Is there still tubulin and HSP70 pulled down with endogenous PP1 and R15? If so, and no actin can be detected, then how does one interpret the physiological significance of the proposed G-actin interaction*.

As noted above, actin is detected in the endogenous complex and given the focus of this and the accompanying paper on actin’s role in eIF2α dephosphorylation ([4], *eLife)*, we consider the study of the other proteins noted in the PPP1R15 pull-downs to be better suited to future efforts.

2) Related to this point of stoichiometry, if there is a stoichiometric amount of G-actin bound to PP1/R15 then what does this say about how tubulin or HSP70 might also be binding to the complex?

The in vitro experiments presented in the accompanying manuscript ([4], *eLife*), and discussed in detail in the revised version of this paper (see below), clearly document a 1:1:1 functional complex of PPP1R15-PP1-actin, which requires no further components for its selective ability to dephosphorylate eIF2α. While this does not exclude an ancillary role for tubulin or HSP70, parsimony justifies the focus on actin here. It is noteworthy however, that PPP1R15 proteins have extensive portions outside their functional core (which engages PP1 and actin and constitutes the specific eIF2α^P^-directed phosphatase). Thus there is nothing in our observations to argue against further component (tubulin, HSP70) joining the complex in high stoichiometry to regulate other aspects of its function.

We have amended the Discussion as follows:

“Our proteomics analysis also identified other potential binding partners of PPP1R15. […] the evidence at hand suggesting actin’s relevance to the core activity of the eIF2α-directed phosphatase justifies the focus on actin.”

*3) The questions related to the binding between the different proteins really needs evidence using purified components, which is the topic of the accompanying paper* ([4], *eLife*). *In isolation it is not easy to judge the potential physiological significance of the data presented in this paper, and the Discussion really needs to make it more clear how the structural data in the other paper help interpret the data in this paper*.

We agree and have extended the Discussion as follows:

“A crystal structure obtained for the binary complex of PPP1R15B and PP1 demonstrated that the N-terminal half of PPP1R15’s functional core […] suggests a mode of engagement of actin by this portion of PPP1R15B that would be sensitive to disruption by cytochalasin, which binds to the same region of G-actin.”

*4) Abstract: the data presented here are more supportive of G-actin being able to stabilize the PP1/R15 complex rather than being an ”activator”*.

We have changed this word in the Abstract.